



# Imprints of evaporation and vegetation type in diurnal temperature variations

Annu Panwar[1], Maik Renner[1], Axel Kleidon[1]

[1]Biospheric Theory and Modeling group, Max Planck Institute for Biogeochemistry, Jena, 07745, Germany

*Correspondence to*: Annu Panwar (apanwar@bgc-jena.mpg.de)

Abstract. Diurnal temperature variations are strongly shaped by the absorption of solar radiation, but evaporation, or the latent heat flux, also plays an important role. Generally, evaporation cools. Its relation to diurnal temperature variations, however, is unclear. This study investigates the diurnal response of surface and air temperatures to evaporation for different vegetation types. We used the warming rate of temperature to absorbed solar radiation in the morning under clear-sky conditions and evaluated how the warming rates change for different evaporative fractions. Results for 51 FLUXNET sites show that the diurnal variation of air temperature carries very weak imprints of evaporation across all vegetation types. However, surface temperature warming rates of short vegetation decrease significantly by $\sim23 \times 10^{-3}$ K/W m$^{-2}$ from dry to wet conditions. Contrarily, warming rates of surface and air temperatures are similar at forest sites and carry literally no imprints of evaporation. We explain these contrasting patterns with a surface energy balance model. The model reveals a strong sensitivity of the warming rates to evaporative fraction and aerodynamic conductance. However, for forests the sensitivity to evaporative fraction is strongly reduced by 74 % due to their large aerodynamic conductance. The remaining imprint is reduced further by $\sim$ 50% through their enhanced aerodynamic conductance under dry conditions. Our model then compares the individual contributions of solar radiation, evaporation and vegetation types in shaping the diurnal temperature range. These findings have implications for the interpretation of land-atmosphere interactions and the influences of water limitation and vegetation on diurnal temperatures, which is of key importance for ecological functioning. We conclude that diurnal temperature variations may be useful to predict evaporation for short vegetation. In forests, however, the diurnal variations in temperatures are mainly governed by their aerodynamic properties resulting in no imprint of evaporation in diurnal temperature variations.

## 1 Introduction

Temperature is one of the most widely monitored variables in meteorology. Besides being important for our day-to-day activities, temperature serves as a primary attribute for the understanding of Earth system processes. The diurnal variation of temperature is considered informative in climate science, as described by the diurnal temperature range (DTR), which is basically the difference between daily maximum and minimum temperatures. Information on the diurnal temperature range has facilitated a broad spectrum of research including agriculture, health welfare, climate change and ecological studies.



Over land the diurnal variation of temperature is mainly driven by the solar energy input (Bristow and
Campbell, 1984). Liu et al., (2004) shows a high correlation of 0.88 between the annual solar radiation
and DTR in China. Likewise, Makowski et al., (2009) found their annual correlation to be 0.87 for
Europe. Their obvious and still intricate association is also important in determining the influence of
solar dimming and brightening on diurnal temperature variations (Wang and Dickinson, 2013; Wild,

45 2005).

Solar radiation is the dominant, but not the only, factor shaping the diurnal temperature variation.
Available energy at the surface is partitioned into latent and sensible heat flux. A higher latent heat flux
signifies higher evaporation, which reduces the temperature through evaporative cooling, an effect that
can be seen in global climate model sensitivities to land evaporation (Shukla and Mintz, 1982).
Another climate model-based analysis (Mearns et al., 1995) shows that differences in evaporation
explain 52 % of the variance in DTR in the summer season for the USA. Similarly, climate model
simulations also show the high sensitivity of DTR to evaporation especially in the summer season
when evaporation is not energy limited (Lindvall and Svensson, 2015). Consequently, methods to
estimate evaporation use air temperature (Blaney and Cridlle, 1950; Hargreaves and Samani, 1985;
Thornthwaite, 1948) and remotely sensed surface temperature (Anderson et al., 2012; Boegh et al.,
2002; Jackson et al., 1999; Kustas and Norman, 1999; Price, 1982; Su et al., 2007). Most of the surface
energy balance based estimates of evaporation use DTR as an input (Baier and Robertson, 1965;
Vinukollu et al., 2011; Yao et al., 2013).
Clouds, precipitation, and atmospheric composition are also important factors that determine DTR (Dai
et al., 1999; Stenchikov and Robock, 1995). One can exclude their contribution to some extent by
considering only clear sky days to more clearly identify the role of evaporation on DTR. Furthermore,
the partitioning of the turbulent heat fluxes into sensible and latent heat is also affected by vegetation
type. Taller vegetation has a higher aerodynamic conductance that facilitate mass and heat exchange
between land and atmosphere (Jarvis and  McNaughton, 1986). The greater conductance in forests
reduces their DTR by reducing their maximum temperature (Bevan et al., 2014; Gallo, 1996; Jackson
and Forster, 2010). Few studies captured the impact of aerodynamic properties of vegetation on
temperature, for example, in terms of the decomposed temperature metric theory (Juang et al., 2007;
Luyssaert et al., 2014) and the theory of intrinsic biophysical mechanism (Lee et al., 2011; Zhao et al.,
2014). Generally, the lower temperatures of forests are associated with their mean evaporative
environment, although this may be affected by periods of dry and wet conditions.
In this study we investigate how the diurnal variation in surface and air temperature responds to
changes in evaporative conditions in different vegetation types. Clearly, DTR is not independent of
solar radiation, which is why we develop an alternative characteristic, the warming rate that eliminates
the contribution of solar radiation. To illustrate this, observed diurnal air and surface temperatures are
plotted against absorbed solar radiation for a cropland and forest site in Figure 1. The diurnal evolution
of temperature is mainly governed by the absorbed solar radiation ($R_s$); this is discernible from the
linear increase in the morning (20 W m$^{-2}$ ≤ $R_s$ ≤ $R_{s,max}$), as described by the slope. This dependence is





accounted for by what we refer to as the warming rate, the increase in temperature due to a unit
increase in the absorbed solar radiation, expressed as the derivative $dT_a/dR_s$ for air temperature and
$dT_s/dR_s$ for surface temperature with units of K/W m$^{-2}$. One can approximate the warming rate by the
ratio of DTR to maximum solar radiation, so that the warming rate can be seen as an efficient
characteristic that captures effects on DTR that are not caused by solar radiation. In this study, we use
linear regressions of observed data from the morning to noon to calculate warming rates.


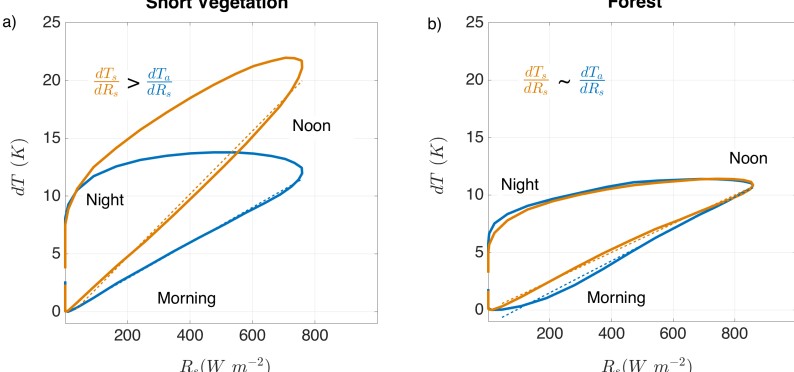

**Figure 1** Mean diurnal hysteresis formed by plotting the diurnal temperature anomaly (y-axis) against
absorbed solar radiation (x-axis) for summer clear sky days. Surface temperature is depicted in orange
and air temperature in blue. (a) A short vegetation cropland site (US-ARM) in Southern Great Plains
Lamont OK, United States. (b) A forest site (CA-TP4) in Ontario, Canada. The dashed lines are the
linear regression of the observations falling in the morning slope of the hysteresis that corresponds to
the warming rate $(dT/dR_s)$ of air $(T_a)$ and surface temperature $(T_s)$.

The temperature warming rate provides insights on the effects of vegetation on the diurnal variation of
temperatures. Figure 1a shows a greater surface temperature warming compared to air temperature for
a cropland site. Contrarily, the warming rates of the two temperatures are similar for a forest site
(Figure 1b). This indicates the strong coupling of diurnal air and surface temperatures in forests
compared to short vegetation.

Certainly, it is intriguing to find out how evaporation alters this coupling. In our earlier work (Panwar
et al., 2019) we looked at the temperature warming rate for a cropland site in the Southern Great Plains.
We observed that the warming rate of surface temperature decreases from dry (less-evaporative) to wet
(evaporative) conditions but the warming rate of air temperature remained unaffected by evaporation.
Combining the boundary layer information and heat budget expression we explained that the diurnal
variation of air temperature does not contain the imprints of evaporation due to the compensating role
of boundary layer development. If this is a general finding, then the surface temperature warming rate





can be used for estimating evaporation of short vegetation. However, it is also interesting to see how
the evaporative cooling effect competes with the cooling effect by a higher aerodynamic conductance.

In this study, we approach two major questions to advance our understanding of diurnal temperature
variations: a) Do the diurnal variations of surface and air temperature respond to evaporation? and b)
What is the role of the different aerodynamic conductance of vegetation in altering these responses?
Our previous work (Panwar et al., 2019) already shows the stronger imprints of evaporation in diurnal
surface temperature variations. Here we examine the generality of this finding in short vegetation.
Additionally, to understand the role of aerodynamic conductance in modifying these imprints we
analyze data from the taller and more complex vegetation like savanna and forests.

We first present a model based on the surface energy balance to provide an expression for the surface
temperature warming rate and its response to evaporation and aerodynamic conductance (all variables
used are summarized in Table A1). To evaluate our model, we used observations from 51 FLUXNET
sites that include short vegetation, savanna and forests. Surface and air temperature warming rates,
aerodynamic conductances and their response to evaporation are quantified for each site. We then use
these findings with our model to explain and reproduce observed temperature warming rates and their
response to evaporation. The cooling effect of evaporation and its relation to aerodynamic conductance
is quantified for each vegetation type. Combining the warming rates with the information on solar
radiation, we conclude the study by demonstrating the contribution of solar radiation, evaporation and
aerodynamic conductance in shaping the DTR using our observational analysis and model.

**2 Modeling temperature-warming rate**
Surface and air temperatures possess a strong diurnal variation that is driven by the absorbtion of solar
radiation. The amplitude of this variation is also affected by other components of surface energy
balance, among which the partitioning of turbulent heat fluxes into latent and sensible heat is
important. Generally, the surface energy balance is written as

$R_s = R_{l,net} + LE + H + G$  .                           Eq. (1)

Here, $R_s$ is the absorbed solar radiation at the surface, $R_{l,net}$ is the net longwave radiation, $LE$ is the
latent heat flux (with $L$ being the latent heat of vaporization and $E$ the evaporation rate), $H$ is the
sensible heat flux and $G$ is the ground heat flux. For simplification of the surface energy balance we
linearize $R_{l,net}$ using the first order terms, such that $R_{l,net} = R_o + k_r (T_s - T_{ref})$. Here, $R_o$ is the net
radiation at a reference temperature $T_{ref}$. The second term, $k_r = 4 \sigma T_{ref}^3$ is the linearization constant.
Incorporating this simplification of $R_{l,net}$ in Eq. (1), the surface energy balance can be rearranged to
yield an expression for $T_s$,





$$T_s = T_{ref} + \frac{R_s - R_o - LE - H - G}{k_r} \qquad \text{Eq. (2)}$$


The warming rate of surface temperature is obtained by taking the derivative of Eq. 2 with respect to
absorbed solar radiation, $R_s$. The warming rate of surface temperature is given by

$$\frac{dT_s}{dR_s} = \frac{1}{k_r} - \frac{1}{k_r} \frac{d(H + LE)}{dR_s} \qquad \text{Eq. (3)}$$


Since, $R_o$ and $T_{ref}$ are assumed to be constants and do not vary diurnally with $R_s$, they disappear in
Eq. (3). Additionally it is assumed that the diurnal change in $G$ in response to $R_s$ is negligible
$(dG/dR_s \sim 0)$ compared to other components of surface energy balance. This assumption is valid since
we are considering vegetated sites for our study, although we are aware that for non-vegetated surfaces
$G$ can represent a noticeable share of absorbed solar radiation (Clothier et al., 1986; Kustas and
Daughtry, 1990).

We describe the evaporative conditions by the evaporative fraction ($f_e$), the ratio of the latent heat
flux ($LE$) to the total turbulent heat fluxes ($H + LE$). Given this, the term $H + LE$ in Eq. (3) can be
written as $H/(1 - f_e)$. Furthermore, the sensible heat flux can be expressed in terms of the
aerodynamic conductance as $H = c_p \rho \, g_a (T_s - T_a)$, where $c_p$=1005 J/kg K is the specific heat capacity
of air, $\rho = 1.23$ kg m$^{-3}$ is air density and $g_a$ is the aerodynamic conductance. On including these
replacements in Eq. (3) we get an approximation for the surface temperature warming rate

$$\frac{dT_s}{dR_s} = \frac{(1 - f_e) + c_p \rho \, g_a \, (dT_a/dR_s)}{k_r \, (1 - f_e) + c_p \rho \, g_a} \qquad \text{Eq. (4)}$$


where $dT_a/dR_s$ is the air temperature warming rate. We can further simplify this expression by
considering the two terms in the denominator of Eq. (4). Considering $T_{ref} \sim 288$ K, the term
$k_r \, (1 - f_e)$ varies by $\sim 4.87$ Wm$^{-2}$K$^{-1}$ to $\sim 0.54$ Wm$^{-2}$K$^{-1}$ from dry ($f_e \sim 0.1$) to wet ($f_e \sim 0.9$) conditions,
which is much smaller in magnitude compared to the term $c_p \rho \, g_a$ that is $\sim 60$ Wm$^{-2}$K$^{-1}$ for a typical
cropland site ($g_a$ =0.05 m s$^{-1}$) and 250 W m$^{-2}$K$^{-1}$ for a typical forest site ($g_a$ =0.2 m s$^{-1}$). Because of
these magnitudes, the term $k_r \, (1 - f_e)$ can be neglected. This leads to a further simplification of the
warming rate to

$$\frac{dT_s}{dR_s} \approx \frac{(1 - f_e)}{c_p \rho \, g_a} + \frac{dT_a}{dR_s} \qquad \text{Eq. (5)}$$


Eq. (5) shows that morning to noon warming of surface temperature is a function of evaporative
fraction, aerodynamic conductance and also of the warming rate of air temperature.





Finally, the sensitivity of the warming rate to changes in evaporative conditions is obtained by taking
the derivative of Eq. (5) with respect to evaporative fraction ($f_e$). To express these derivatives with
respect to evaporative fraction, we use the apostrophe ($d/\,df_e = '$). Therefore, $dT_s'/dR_s$ and
$dT_a'/dR_s$ represent the change in surface and air temperature warming rates due to a unit change in
evaporative fraction. Similarly, $g_a'$ is the change in aerodynamic conductance from dry to wet
evaporative conditions. We obtain:

$$\frac{dT_s'}{dR_s} = -\frac{1}{c_p \cdot \rho \cdot g_a} - \frac{1-f_e}{c_p \cdot \rho \cdot g_a} \cdot \frac{g_a'}{g_a} + \frac{dT_a'}{dR_s} \qquad \text{Eq. (6)}$$


This model provides two important expressions that we test with observations. The first expression is
the warming rate of surface temperature, described by Eq. (5), which requires the information of the
warming rate of air temperature, aerodynamic conductance and evaporative fraction. On multiplying
these two equations with daily maximum solar radiation shall provide an approximation of DTR that
can also be validated with the observational data. The second expression is the response of the surface
temperature warming rate to evaporation, shown in Eq. (6), which is a negative quantity provided
$dT_a'/dR_s$ is small (or negative). The negative sign means that the surface temperature warming rate
decreases with increase in evaporative fraction. The amplitude of this decrease mainly depends on the
characteristic aerodynamic conductance ($g_a$) of vegetation (the first term on the right hand side of Eq.
6) and also on its relative sensitivity to evaporative fraction ($g_a'/g_a$, the second term on the right hand
side).

**3 Data and method**
We use observations from 51 FLUXNET sites representing different vegetation types. The FLUXNET
data consists of sensible and latent heat fluxes using the standard eddy covariance method and provides
half hourly radiation and meteorological data (Baldocchi et al., 2001). The selected 51 sites contain
data of the surface energy balance components and temperatures for more than four years. To avoid the
effect of energy limitation on evaporation only summer days are considered. Summer is defined here as
days having their daily mean incoming solar radiation at the surface greater than the median of the
annual distribution. This approach standardizes the definition of summer days for sites at different
latitudes and provides the days with comparable solar energy input for the individual sites.

Furthermore, among summer days only clear sky days are considered to avoid the influence of clouds
on temperatures. The process of obtaining summer days already filters out the days with high duration
of cloud covers that result in reduced mean incoming solar radiation. An additional filter to remove
cloudy days is applied that is based on the quantile regression method using surface solar radiation and
potential solar radiation (Renner et al., 2019). This method was applied only from morning to noon, so
that if the day has clouds in the evening, it is still considered as a clear sky day. This does not influence
warming rates since they are calculated only from the morning to noontime variation of temperature.




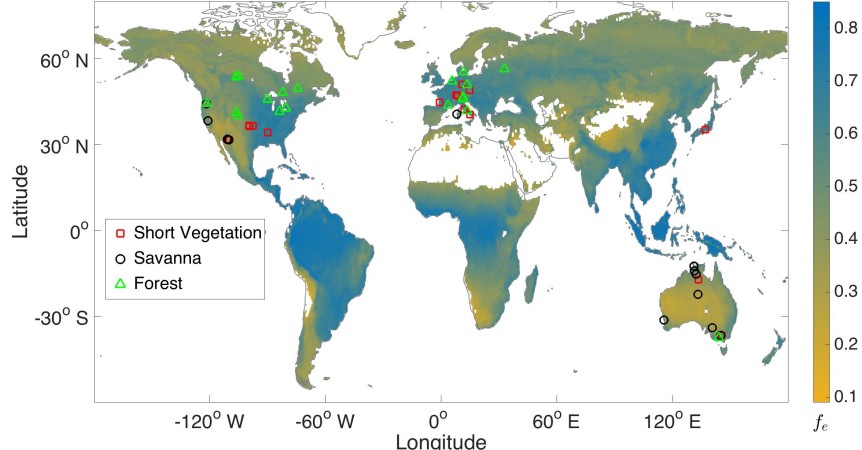


**Figure 2** Geographical locations of FLUXNET sites used in this study. The vegetation type at each site
is shown by the symbols. The color bar shows the mean annual evaporative fraction ($f_e$) derived from
FLUXCOM data (2001 to 2013).

The vegetation type of each site is classified using the International Geosphere-Biosphere Programme
(IGBP) Data and Information System (Loveland and Belward, 1997). The IGBP land cover product is
available at 1 km resolution and was derived from the Advance Very High Resolution Radiometer
(AVHRR). Detailed information of each site with their location, number of days used in the analysis,
land use type and references is provided in the Appendix (Table A2). Vegetation are classified into
three types that is based on their typical vegetation height and coverage, see Table 1. Shorter vegetation
like croplands, grasslands, and shrublands are grouped into the 'short vegetation' type. Savanna
ecosystems are complex with heterogeneous vegetation height, which basically delineates the transition
of short vegetation to forests, and are grouped into the 'savanna' type. All forest types, including
deciduous broadleaf, evergreen broadleaf, evergreen needleleaf and mixed, are grouped in the 'forest'
type.

The geographic location of the selected 51 sites is shown in Figure 2. The color bar represents the
mean annual evaporative fraction derived from FLUXCOM data (Jung et al., 2019; Tramontana et al.,
2016). Selected sites represent a wide range of ecosystems that is ideal for studying the generality of
the response of warming rates to differences in evaporative conditions and vegetation type.

**Table 1.** Land use types of the different sites considered here and their grouping into the short
vegetation, savanna and forest types.

| Vegetation types | Land use type | Number of sites |
|---|---|---|
| Short Vegetation | Cropland | 12 |





| | Grassland | 6 |
|---|---|---|
| | Shrubland | 5 |
| Savanna | Savanna | 4 |
| | Woody Savanna | 5 |
| Forest | Deciduous broadleaf forest | 4 |
| | Evergreen broadleaf forest | 1 |
| | Evergreen needle leaf forest | 9 |
| | Mixed forest | 5 |


The evaporative condition is quantified by evaporative fraction. One of the advantages of evaporative
fraction is its stability for daylight hours such that it can be assumed to be constant over a day
(Shuttleworth et al., 1989). Daily evaporative fraction is obtained by the linear regression of half hourly
morning to noon values of the ratio of the latent heat flux to the total turbulent heat fluxes. Similarly, a
linear regression of half hourly warming rate and evaporative fraction values is used to quantify the
response of the warming rate to evaporative fraction.

We use the term air temperature for the temperature measured above the canopy. Surface temperature
is calculated from the upwelling longwave radiation using the Stefan-Boltzmann law, such that the
surface temperature is the skin temperature of the vegetation. The aerodynamic conductance ($g_a$) is
obtained from the observed frictional velocity ($u_*$) and wind speed ($u$) by $g_a = u_*^2/u$ (see, e.g.,
Verma (1989)). For simplicity, the conductance of heat fluxes and momentum are assumed to be
identical (Mallick et al., 2016).

**4 Results**
4.1 Observational analysis

The primary advantage of warming rate over DTR is its suitability to compare sites with different solar
energy input. This is apparent from Figure 3, where we show the probability density distribution of the
observed daily warming rates of surface (a) and air temperatures (b) for short vegetation, savanna, and
forest. We look at the surface and air temperature warming rates to determine if they carry any
information on vegetation type. In general, the surface temperature warming rate of short vegetation is
larger by almost a factor of two compared to the surface temperature warming rate of forests. Savanna
covers the range in surface temperature warming rates, reflecting their characteristics being positioned
between short vegetation and forests. Hence, the vegetation type clearly affects the surface temperature
warming rate. Surprisingly, this is not true for air temperature warming rates. Short vegetation, savanna
and forests show similar distributions of air temperature warming rate. The air temperature warming
rate of short vegetation is smaller than its surface temperature warming rate. Conversely, in forests, the
magnitudes are similar, indicating the strong coupling between surface and air temperature.

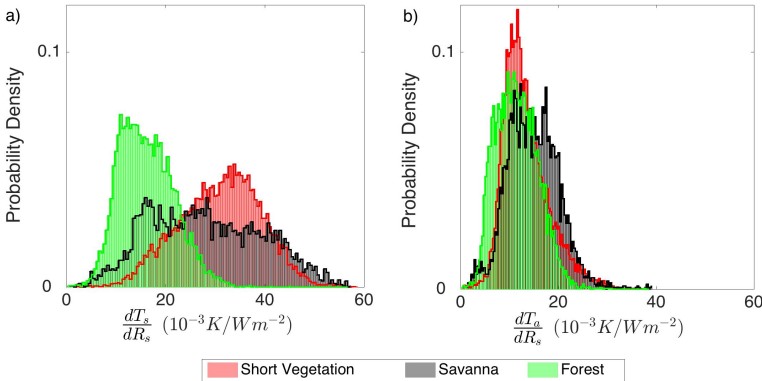

**Figure 3** Probability density distribution of observed (a) surface temperature warming rates and (b) air
temperature warming rates for short vegetation (red), savanna (grey) and forest (green).

Site-specific information on warming rates is provided in Figure A1 in the Appendix. Within the short
vegetation type, grassland and shrubland sites show much greater surface temperature warming rates
than air temperature warming rates. This distinction could be attributed to the site-specific evaporative
conditions. Evaporative conditions at some sites have a certain general tendency. For instance, most of
the shrubland sites are drier, cropland sites are generally wetter and forest sites show intermediate
evaporative fractions. Such an uneven distribution of evaporative conditions could impact the warming
rates, such that it is higher for dry and lower for wet sites. On the other hand, despite these differences
in the mean, the sites contain days with a good range of evaporative fractions (see Figure A2 in
Appendix). The range of evaporative fractions is important to calculate the sensitivity of warming rates
to evaporative fraction.

Next, we quantify the response of surface and air temperature warming rates to changes in evaporative
fraction from dry to wet conditions. The warming rate response to evaporative fraction is obtained from
the linear regression of daily warming rates to daily evaporative fractions for each site. Figure 4 shows
the mean response of the surface (orange) and air (blue) temperature warming rate to evaporative
fraction for short vegetation, savanna and forest. For site-specific responses, see Figure A2 in the
Appendix. It is noticeable that regardless of the magnitudes of the warming rates and different mean
evaporative conditions, the response of warming rates to evaporative fraction is almost consistent for
the different vegetation types. For instance, the surface temperature warming rate of short vegetation
shows a consistent decrease of $\sim 23 \times 10^{-3}$ K/W m$^{-2}$ from dry to wet days. However, the air temperature
warming rate decreases only by $\sim 5 \times 10^{-3}$ K/W m$^{-2}$. In our earlier work, similar responses were
observed for a cropland site (See Appendix, site 8). We find a similarly weak response for savanna and
forests. In savanna, the surface temperature warming rate still decreases by $\sim 12 \times 10^{-3}$ K/W m$^{-2}$ from dry
to wet conditions, but the air temperature warming rate remains almost the same. In forests, both,



surface and air temperature warming rates, show very weak to almost no response to evaporative
fraction, although with some variations as reflected by the error bars.

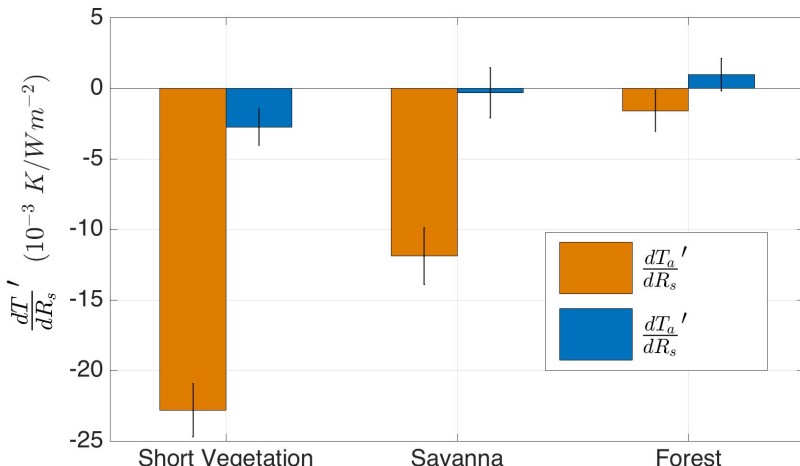


**Figure 4** Bar plots of the observed mean response of surface ($dT_s' / dR_s$) and air ($dT_a' / dR_s$)
temperature warming rates to changes in evaporative fraction for short vegetation, savanna and forests.
The error bars represent the standard error in the mean of all sites in the respective type.

Besides evaporation, the aerodynamic conductance also influences the diurnal variation of temperature.
The aerodynamic conductance governs the ventilation of energy and mass from the surface to the
atmosphere (Thom, 1972). Figure 5 shows the mean aerodynamic conductances for the vegetation
types. The mean aerodynamic conductance is usually a characteristic of vegetation height but
variations might occur due to changing evaporative conditions. In general, it is observed that the
aerodynamic conductance of short vegetation is much lower than the aerodynamic conductance of
forest. Savannas show relatively higher aerodynamic conductances compared to short vegetation. Some
woody savannas have comparable aerodynamic conductances to forests. Forests have generally high
aerodynamic conductances.

In addition to the mean aerodynamic conductance we also observed its response to evaporative
fraction. The change in aerodynamic conductance due to the change in evaporative fraction is denoted
by $g_a'$ that is derived from the linear regression of their observed daily values. The negative sign of
$g_a'$ reflects the decrease in $g_a$ from dry to wet days so that the aerodynamic conductance is enhanced
on days with low evaporative fraction. Site-specific values of $g_a$ and $g_a'$ are provided in Figure A3 in
the Appendix. For all vegetation types the aerodynamic conductance increases on dry days. For short
vegetation this increase is about ~50 % such that the characteristic aerodynamic conductance of 0.041
m s$^{-1}$ increases to 0.055 m s$^{-1}$ on dry days. For forests, the aerodynamic conductance for most of the
sites increases by ~100% on dry days. This enhancement becomes considerably important for forests
because it increases their already large aerodynamic conductance, for instance from 0.12 m s$^{-1}$ to 0.24





m s$^{-1}$. However, the main distinction between forest and short vegetation remains their mean
aerodynamic conductance whereas the enhanced aerodynamic conductance is just a secondary factor
whose impact on warming rate is further analyzed using our model.

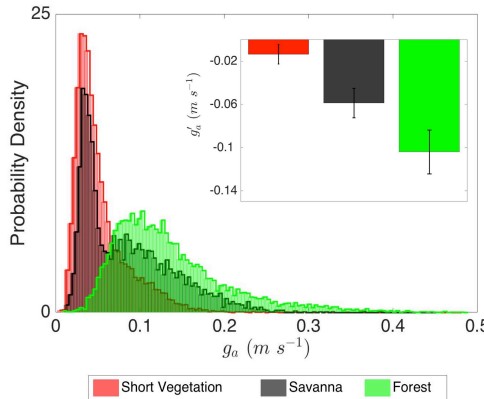


**Figure 5** Probability density distribution of the aerodynamic conductance ($g_a$) derived from
observations at the 51 FLUXNET sites for short vegetation, savanna and forests. The inset plot shows
the mean sensitivity of aerodynamic conductance to evaporative fraction ($f_e$) for the three types, $g'_a$.
The error bar represents the standard error in the mean regression of $g_a$ and $f_e$.

**Table 3.** First quartile (Q1), median and third quartile (Q3) for the observed distribution of $dT_s/dR_s$,
$dT_a/dR_s$ and $g_a$ for short vegetation, savanna and forest.

| Vegetation | | $dT_s/dR_s$ (10$^{-3}$ K/W m$^{-2}$) | $dT_a/dR_s$ (10$^{-3}$ K/W m$^{-2}$) | $g_a$ (m s$^{-1}$) |
|---|---|---|---|---|
| Short Vegetation | Q1 | 25.1 | 9.9 | 0.029 |
| | Median | 31.4 | 12.3 | 0.041 |
| | Q3 | 36.7 | 15.7 | 0.063 |
| Savanna | Q1 | 18.6 | 10.9 | 0.037 |
| | Median | 27.1 | 14.4 | 0.068 |
| | Q3 | 36.8 | 18.1 | 0.115 |
| Forest | Q1 | 11.8 | 8.1 | 0.085 |
| | Median | 15.5 | 11.1 | 0.118 |
| | Q3 | 19.7 | 14.3 | 0.164 |


To summarize our observational analysis, we show that the diurnal variation of surface temperature of
short vegetation carries stronger imprints of evaporative conditions compared to the diurnal variation
of air temperature. In forests, the diurnal variations of both, surface and air temperature do not respond
to evaporative conditions. Observations also demonstrate characteristic high aerodynamic
conductances of forests compared to short vegetation. Additionally, we showed an enhancement of





aerodynamic conductance on dry days that is relatively stronger for forests compared to short
vegetation.

To explain these findings we hypothesize that the high aerodynamic conductance of forest and its
enhancement on dry conditions lowers the diurnal warming of surface temperature. Consequently, the
warming rates of surface temperature of forests are less sensitive to evaporation. Our hypothesis is
based on the observational based findings and the interpretation of the model equations where one can
determine the contribution of $g_a$, $f_e$ and $g'_a$ in shaping the warming rates. This can already be
anticipated from Eq. 6, evaluated with the median values provided in Table 3. The first term on the
right-hand side of Eq. 6 is about -21 x $10^{-3}$ K/(W m$^{-2}$) for short vegetation, but only -7 x $10^{-3}$ K/(W m$^{-2}$)
for forests, similar to what is shown in Figure 4. In the next section we verify our hypothesis using the
modeled expression for surface temperature warming rate and its response to evaporative conditions.
Along with the model evaluation we quantify the contribution of aerodynamic conductance and its
enhancement in compensating the imprints of evaporation on warming rates for surface temperature for
short vegetation, savannas and forests.

4.2 Model interpretation
In this section we estimate the surface temperature warming rate and its response to evaporative
fraction using our model, which is then compared to observations. Then we use the model to quantify
the contribution of evaporative fraction and of aerodynamic conductance to the diurnal temperature
range.

To model the warming rate we use Eq. (5), in which the vegetation type is captured by $g_a$, and
evaporative conditions by $f_e$. The model sensitivity of surface temperature warming rate to evaporative
fraction and aerodynamic conductance is shown in Figure 5a. The model shows a stronger gradient of
the warming rate with evaporative fraction for low aerodynamic conductances. As in the observations
warming rates for low aerodynamic conductances are greater compared to high aerodynamic
conductances. Observed warming rates for short vegetation, savannas and forests are also plotted in
Figure 5a, using their mean evaporative fractions and aerodynamic conductances, respectively. Note
that both, evaporative fraction and aerodynamic conductance, can vary. This implies that the position
of the sites can change vertically and somewhat horizontally with changes in evaporative conditions.

Almost all of the short vegetation sites have low aerodynamic conductances where the warming rate
increases with decreasing evaporative fraction. Contrarily, the forest sites show no such strong
variation in warming rate. This is consistent with the study by Diak and Whipple (1993), who showed
a similar dependency of the diurnal range of surface temperature on the Bowen ratio and surface
roughness length using a boundary layer model simulation. Our model can capture these patterns solely
with surface energy balance information and requires no information of the boundary layer. This





indicates that the diurnal variation in surface temperature is dominantly governed by the exchange at
the surface, particularly aerodynamic conductance and evaporative fraction.

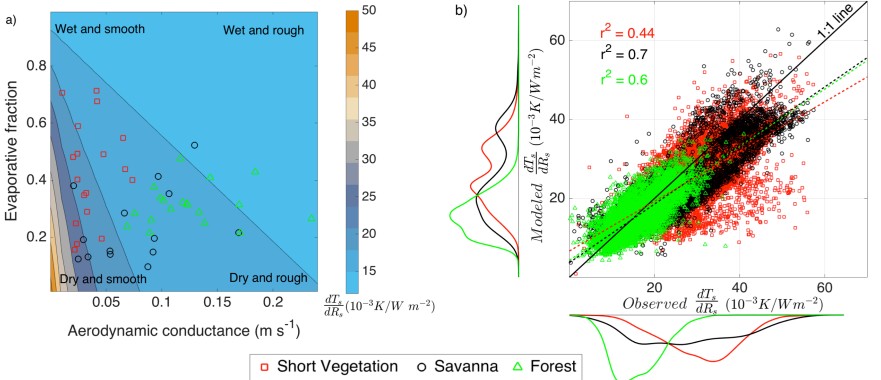


**Figure 6** a) Modeled surface temperature warming rate ($dT_s/dR_s$) for different aerodynamic
conductances ($g_a$, x-axis) and evaporative conditions ($f_e$, y-axis). The color bar shows the magnitude
of the warming rate. The symbols correspond to the different sites, using their mean aerodynamic
conductance and evaporative fraction. b) Modeled versus observed daily warming rates, $dT_s/dR_s$, for
each site for the three vegetation types. The histograms show the distribution and spread. The
coefficient of determination ($r^2$) is depicted for the linear regressions (dashed lines).

We next tested the model by estimating the daily surface temperature warming rate for each site using
Eq. (5) from daily values of observed $f_e$, $g_a$ and $dT_a/dR_s$. Since $dT_a/dR_s$ is similar for all sites, the
diurnal variation of air temperature does not seem to depend on the diurnal variation of surface
temperature, and vice versa. Figure 6b shows the comparison of the modeled surface temperature
warming rates to those derived from observations. The model performs very well for all sites for the
given information. The coefficient of determination ($r^2$) is also high for savanna and forests, pointing at
the functionality of our model for complex and taller vegetation. However, short vegetation shows
slightly weaker $r^2$ because our model underestimates the surface temperature warming rate at a few
short vegetation sites. We speculate that these are the sites with non-vegetated surfaces where the
ground heat flux contribution to diurnal surface temperature variations is significant (Saltzman and
Pollack, 1977) which is currently neglected in our model.

It is apparent from Figure 6 that the response of the surface temperature warming rate to evaporative
fraction is predominantly governed by the aerodynamic conductance. The expression for $dT_s{}'/dR_s$ in
Eq. (6) quantifies this. Note that here we do not assume a constant aerodynamic conductance since $g_a$
in the observation is enhanced on dry days. Our model reproduces the response of the warming rates to
evaporative conditions ($r^2 = 0.6$) for all types, Figure 7a. Additionally, it even captures the ranges in
$dT_s{}'/dR_s$ for the specific vegetation types. Certain deviations exist because there are some biases in
the number of wet and dry days in the observations that is reflected in the horizontal error bars. The





other possible root for a bias is the absence of a clear relation between $g_a$ and $f_e$ at some sites, these
sites are indicated in lighter shades.


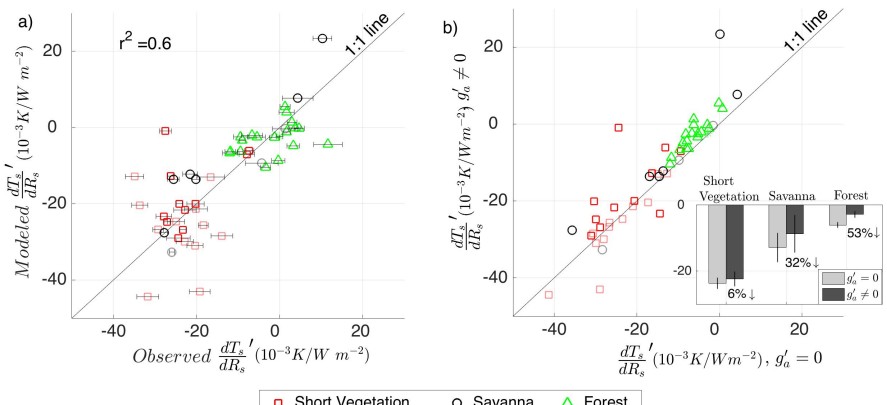



**Figure 7** a) Model evaluation of the response of surface temperature warming rates to evaporative
conditions ($dT_s{'}/dR_s$) with those derived from observations for each site. b) Comparison of modeled
$dT_s{'}/dR_s$ for two cases: the first case assumes the aerodynamic conductance to be insensitive to
evaporative fraction ($g'_a$=0, x-axis). The second case includes the sensitivity of aerodynamic
conductance to evaporative fraction ($g'_a \neq 0$, in y-axis). The inset bar plot compares the mean
contribution for the two cases with the error bars representing standard error of the mean. Sites with
non-significant $g'_a$ are marked by lighter shades.

Figure 7b shows the contribution of the enhanced aerodynamic contribution on drier days in
compensating the response of warming rates to evaporative conditions. For this, we compare the
modeled $dT_s{'}/dR_s$ with and without the inclusion of enhanced aerodynamic conductance term (the
second term on the right-hand side of Eq. 6), such that $dT_s{'}/dR_s$ when $g'_a = 0$ only captures the
contribution of mean aerodynamic conductance and $dT_s{'}/dR_s$ when $g'_a \neq 0$ additionally shows the
contribution of the enhanced aerodynamic conductance on drier days. For the comparison of the two
cases it is important to recognize that the more negative the values of $dT_s{'}/dR_s$, the stronger the
imprint of evaporation is in the diurnal variation of temperature.

In general, for most of the sites the enhanced aerodynamic conductance plays a small, but noticeable
role in weakening the response of the warming rate to evaporative fraction. This is evident since the
data points lie above the 1:1 line and tend to be less negative for the case when $g'_a \neq 0$. This effect is,
however, more consistent for forests compared to short vegetation and savannas (see the inset bar plot
which summarizes the mean $dT_s{'}/dR_s$ for two cases). For short vegetation sites, $dT_s{'}/dR_s$ decreases
only by 6% when $g'_a \neq 0$ is considered. For savannas, the decrease is 32 %, and it is highest with 53%



for forests. This suggests that along with the inherent high aerodynamic conductance of forests, its
enhanced aerodynamic conductance is also responsible for the absence of evaporation imprints in the
diurnal variation of temperature. The higher aerodynamic conductance of forests is responsible for
reducing 74 % of the imprints of evaporation in diurnal surface temperature when compared to the
short vegetation.

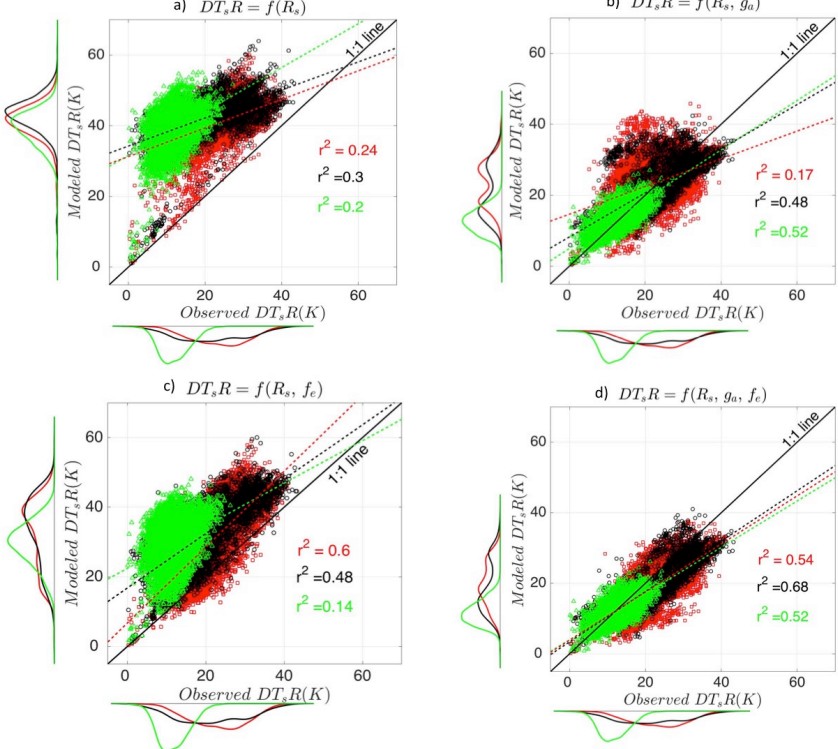


**Figure 8** Comparison of model estimates of the diurnal surface temperature range (DT$_s$R) for short
vegetation, savanna and forests with observations for four scenarios: a) DT$_s$R is only a function of solar
radiation, b) DT$_s$R is a function of solar radiation and aerodynamic conductance, c) DT$_s$R is a function
of solar radiation and evaporative fraction, d) DT$_s$R is a function of solar radiation, aerodynamic
conductance and evaporative fraction. Dashed lines show the linear regression between model and
observation.

We next link our model for surface temperature warming rates back to the diurnal variation in surface
temperature. To understand how solar radiation, the aerodynamic conductivity of the different
vegetation types and evaporative fraction individually influence the diurnal variation in temperature,
we can obtain the diurnal surface temperature range (DT$_s$R) by multiplying the expression for warming
rate given by Eq. 5 with the daily maximum in absorbed solar radiation. To quantify the sensitivity of
DT$_s$R to its three main contributors, we considered four cases. In first case, we assume that the diurnal





variation in surface temperature is solely driven by solar radiation, such that there is no evaporation
($f_e = 0$) and the surface has no vegetation, represented by a very low aerodynamic conductance of
$g_a = 0.02$. Figure 8a shows that in this scenario, $DT_sR$ is overestimated for all vegetation types with
poor $r^2 \leq 0.3$. This greater warming indicates that vegetation and evaporation cools surface
temperatures. In second case we added the information on aerodynamic conductance of each vegetation
type along with solar radiation (Figure 8b). The $DT_sR$ estimates for forests ($r^2 = 0.54$) and to some extent
for savanna ($r^2 = 0.48$) is considerably improved, but not for short vegetation ($r^2 = 0.17$). Nevertheless, in
this case $DT_sR$ is much cooler and closer to the observed values, indicating the importance of
aerodynamic conductance in cooling the diurnal temperature. Aerodynamic conductance alone does not
explain the scatter in $DT_sR$ in short vegetation. In third case we kept the information on daily
evaporative conditions but assumed a very low $g_a = 0.02$ (Figure 8c). Contrarily to Figure 8b, $DT_sR$ in
short vegetation is captured much better ($r^2 = 0.6$), but the magnitudes are overestimated. Similar to
short vegetation, $DT_sR$ is also overestimated for savanna and forest. For forests, $r^2$ is very low, because
the aerodynamic conductance is the key property affecting $DT_sR$. Finally; we added the information on
all the components of the model, solar radiation, aerodynamic conductance and evaporative fractions
(Figure 8d). Compared to the previous three cases the estimates are much closer to the observation with
a good $r^2$ for all vegetation types. This sensitivity analysis shows that vegetation type and evaporation
play significant roles in driving the diurnal variation in surface temperature. Evaporation is important
to capture the spread whereas aerodynamic conductance is important to capture the magnitudes of
diurnal variation of surface temperature, particularly for forest sites.
**6 Discussion**
We demonstrate a robust way of characterizing the diurnal variation of temperature using their morning
to noon warming rates, which are derived from the half hourly temperatures and solar radiation. The
warming rate is suitable for the comparison of locations with different solar energy input whereas other
metrics like diurnal temperature range depends on solar radiation (Makowski et al., 2009).
Consequently, temperature warming rates for specific vegetation types are comparable for sites at
different geographic locations. Our surface energy balance model can reproduce the warming rate and
shows the physical significance of evaporation and aerodynamic conductance. The model can capture
the diurnal variation of temperature quite well. These approximations can further be improved by a
more detailed formulation of net longwave radiation (which could, for instance, include optical
properties of the atmosphere) and the ground heat flux. Warming rates are also sensitive to clouds and
might not capture the information of evaporation and vegetation on cloudy days. Also, we did not
provide a way to calculate warming rates of air temperature. These could represent topics for future
research.
One of the main findings of our study is the different response of diurnal surface and air temperature to
evaporation. The air temperature warming rate does not contain any imprints of evaporation whereas,
for short vegetation, the surface temperature warming rate decreases strongly with evaporative fraction.
This finding is consistent with our previous work where we explained the role of boundary layer in



compensating imprints of evaporative conditions in the diurnal variation of air temperature. We found
that the diurnal variation of air temperature is similar for all vegetation types irrespective of their
aerodynamic conductance and evaporative conditions. We anticipate that our hypothesis of the
compensating effect of boundary layer might also be true for forests, but this would need further
research.

The notion that diurnal surface and air temperature variations respond differently to evaporation should
be considered when developing air temperature products from remotely sensed surface temperature
(Cresswell et al., 1999; Fu et al., 2011; Hengl et al., 2012; Jang et al., 2004; Kilibarda et al., 2014; Zhu
et al., 2013). Typically, these products are primarily based on the assumption that surface temperature
is proxy of air temperature. Generally, these approaches overestimate daytime air temperature (Oyler et
al., 2016; Zhang et al., 2011). This finding is consistent with our results, which show a greater
variation of surface temperature depending on vegetation type and evaporative fraction (cf. Eq. 5).

Our study shows that surface and air temperature warming rates are similar in forests, which indicates
the strong coupling between the two temperatures. This finding is in agreement with the previous study
by Li et al., 2015 and Mildrexler et al., 2011 , where evaporative cooling and high aerodynamic
conductance of forests were identified as the responsible factors for the strong coupling of surface and
air temperature. However, we also show that this coupling remains persistent irrespective of the
evaporative conditions of the forest. Using our model and observations we show that the aerodynamic
conductance of forest increases on dry days resulting in reduced warming of surface temperature and
hence its stronger coupling to air temperature. These findings complement the recent studies on the
convector effect where its role in lowering the surface temperature for a semi-arid forest is discussed
(Banerjee et al., 2017, 2018; Brugger et al., 2019; Kröniger et al., 2018; Rotenberg and Yakir, 2010).
Our demonstration of enhanced aerodynamic conductance on dry days is similar to what these authors
describe as the convector effect.

Unlike forests, the surface temperature warming rate in short vegetation responds strongly to changes
in evaporative conditions. In observations, the warming rate decreases by ~23 x10$^{-3}$ K/W m$^{-2}$ from dry
to wet days. In general, this decrease is comparable for all the short vegetation sites and we anticipate
that some spread is due to their somewhat different aerodynamic properties. Another source of
ambiguity is the unequal distribution of days of different evaporative fractions which also influences
the computation of $dT_s'/dR_s$ and $g_a'$. This constraint requires longer time series of observations to
obtain a greater sampling range of dry and wet days. Overall, our results nevertheless show that the
surface temperature warming rate is a promising indicator of evaporative fraction, especially for short
vegetation.

The other implication of our study is a better physical understanding of the processes that govern the
diurnal temperature range. Our model is capable of capturing the contribution of solar radiation,
vegetation and evaporation in shaping DTR. We show that the aerodynamic conductance of vegetation



is the key-cooling operator whereas evaporation explains the spread in DTR. These findings are
important when interpreting DTR for different ecosystems.
**7 Conclusions**
Temperature and evaporation are among the foremost-discussed variables in hydrology and climate
science. Our study contributes information on the relationship between diurnal temperature variations,
evaporative conditions, and vegetation. To measure the diurnal variation, we introduce the morning to
noontime warming rate of temperature.  This rate has advantages over mean, maximum or minimum
temperatures for conducting a multisite analysis because it removes the effect of solar radiation. We
demonstrated that the warming rate and its response to evaporation is reproducible from the
simplification of surface energy balance. In doing so, we can address the two major questions that we
formulated in the introduction.  First, our observational analysis shows no imprints of evaporation in
the air temperature warming rate across vegetation types. However, the surface temperature response to
evaporation is rather vegetation dependent, being stronger for short vegetation and absent in forests.
These findings provide insights for the second question about the role of aerodynamic conductance.
We showed that the aerodynamic conductance is very important in reducing the diurnal variation of
surface temperature. It is mostly the high aerodynamic conductance of forests, which compensates their
response to evaporative fraction. In addition, the aerodynamic conductance in itself is sensitive to
evaporative conditions. Using observational and model-reproduced findings we demonstrate that along
with the high aerodynamic conductance of forests their aerodynamic conductance roughly doubles on
dry days. The higher aerodynamic conductance results in more efficient transport of heat from the
surface to the atmosphere and compensates for the diurnal rise in surface temperature, which is
reflected in their lower surface temperature warming rate.
To conclude, our results imply that diurnal temperature variations can be understood and predicted by
relatively few factors, solar radiation, aerodynamic conductance and evaporative fraction. Surprisingly,
diurnal air temperature carries little information of vegetation type and evaporative conditions of the
land surface, while surface temperatures carry a stronger imprint of evaporation, but only for short
vegetation.
**Data availability:** For the spatial plot of evaporative fraction we used the FLUXCOM monthly data of
sensible and latent heat fluxes. The FLUXCOM data is available at http://www.fluxcom.org/. For
observational analysis we used FLUXNET data for 52 sites. More description of each site is provided
in the Appendix. For FLUXNET data please see the link https://fluxnet.fluxdata.org/.
**Author contribution:** All authors conceived the study. AP analysed data, AK derived the energy-
balance model that is further developed by AP. MR provided classification of cloud-free conditions.
All authors interpreted results. AP wrote the manuscript with input of MR and AK.



**Competing interests:** The authors declare that they have no conflict of interest.

**Acknowledgements:** Annu Panwar shows her sincere thanks for the stipend from the International
Max Planck Research School for Global Biogeochemical Cycles (IMPRS-gBGC) for performing this
research.

**Reference:**Anderson, M. C., Allen, R. G., Morse, A. and Kustas, W. P.: Use of Landsat thermal
imagery in monitoring evapotranspiration and managing water resources, Remote Sensing of
Environment, 122, 50–65, doi:10.1016/j.rse.2011.08.025, 2012.
Baier, W. and Robertson, Geo. W.: Estimation of latent evaporation from simple weather observations,
Can. J. Plant Sci., 45(3), 276–284, doi:10.4141/cjps65-051, 1965.
Baldocchi, D., Falge, E., Gu, L., Olson, R., Hollinger, D., Running, S., Anthoni, P., Bernhofer, C.,
Davis, K., Evans, R., Fuentes, J., Goldstein, A., Katul, G., Law, B., Lee, X., Malhi, Y., Meyers, T.,
Munger, W., Oechel, W., Paw, K. T., Pilegaard, K., Schmid, H. P., Valentini, R., Verma, S., Vesala,
T., Wilson, K. and Wofsy, S.: FLUXNET: A New Tool to Study the Temporal and Spatial Variability
of Ecosystem–Scale Carbon Dioxide, Water Vapor, and Energy Flux Densities, Bulletin of the
American Meteorological Society, 82(11), 2415–2434, doi:10.1175/1520-
0477(2001)082<2415:FANTTS>2.3.CO;2, 2001.
Banerjee, T., De Roo, F. and Mauder, M.: Explaining the convector effect in canopy turbulence by
means of large-eddy simulation, Hydrol. Earth Syst. Sci., 21(6), 2987–3000, doi:10.5194/hess-21-
602    2987-2017, 2017.

Banerjee, T., Brugger, P., De Roo, F., Kröniger, K., Yakir, D., Rotenberg, E. and Mauder, M.:
Turbulent transport of energy across a forest and a semiarid shrubland, Atmospheric Chemistry and
Physics, 18(13), 10025–10038, doi:10.5194/acp-18-10025-2018, 2018.
Bevan, S. L., Los, S. O. and North, P. R. J.: Response of vegetation to the 2003 European drought was
mitigated by height, Biogeosciences, 11(11), 2897–2908, doi:10.5194/bg-11-2897-2014, 2014.
Blaney, H. F. and Cridlle, W. D.: Blaney HF, Criddle WD. 1950. Determining Water Requirements in
Irrigated Areas from Climatological Irrigation Data. Technical Paper No. 96, US Department of
Agriculture, Soil Conservation Service, Washington, D.C., 48 pp, , 48 pp, 1950.
Boegh, E., Soegaard, H. and Thomsen, A.: Evaluating evapotranspiration rates and surface conditions
using Landsat TM to estimate atmospheric resistance and surface resistance, Remote Sensing of
Environment, 79(2–3), 329–343, doi:10.1016/S0034-4257(01)00283-8, 2002.
Bristow, K. L. and Campbell, G. S.: On the relationship between incoming solar radiation and daily
maximum and minimum temperature, Agricultural and Forest Meteorology, 31(2), 159–166,
doi:10.1016/0168-1923(84)90017-0, 1984.
Brugger, P., De Roo, F., Kröniger, K., Rotenberg, E., Tatarinov, F., Yakir, D., Zeeman, M. and
Mauder, M.: Contrasting turbulent transport regimes explain cooling effect in a semi-arid forest
compared to surrounding shrubland, Agricultural and Forest Meteorology, 269–270, 19–27,
doi:10.1016/j.agrformet.2019.01.041, 2019.
Clothier, B. E., Clawson, K. L., Pinter, P. J., Moran, M. S., Reginato, R. J. and Jackson, R. D.:
Estimation of soil heat flux from net radiation during the growth of alfalfa, Agricultural and Forest
Meteorology, 37(4), 319–329, doi:10.1016/0168-1923(86)90069-9, 1986.
Cresswell, M. P., Morse, A. P., Thomson, M. C. and Connor, S. J.: Estimating surface air temperatures,
from Meteosat land surface temperatures, using an empirical solar zenith angle model, International
Journal of Remote Sensing, 20(6), 1125–1132, doi:10.1080/014311699212885, 1999.



Dai, A., Trenberth, K. E. and Karl, T. R.: Effects of Clouds, Soil Moisture, Precipitation, and Water
Vapor on Diurnal Temperature Range, JOURNAL OF CLIMATE, 12, 23, 1999.
Diak, G. R. and Whipple, M. S.: Improvements to models and methods for evaluating the land-surface
energy balance and 'effective' roughness using radiosonde reports and satellite-measured 'skin'
temperature data, Agricultural and Forest Meteorology, 63(3–4), 189–218, doi:10.1016/0168-
1923(93)90060-U, 1993.
Fu, G., Shen, Z., Zhang, X., Shi, P., Zhang, Y. and Wu, J.: Estimating air temperature of an alpine
meadow on the Northern Tibetan Plateau using MODIS land surface temperature, Acta Ecologica
Sinica, 31(1), 8–13, doi:10.1016/j.chnaes.2010.11.002, 2011.
Gallo, K. P.: The Influence of Land Use/Land Cover on Climatological Values of the Diurnal
Temperature Range, JOURNAL OF CLIMATE, 9, 2941–2944, 1996.
Hargreaves, G. H. and Samani, Z. A.: Reference Crop Evapotranspiration from Temperature, Applied
Engineering in Agriculture, 1(2), 96–99, doi:10.13031/2013.26773, 1985.
Hengl, T., Heuvelink, G. B. M., Perčec Tadić, M. and Pebesma, E. J.: Spatio-temporal prediction of
daily temperatures using time-series of MODIS LST images, Theor Appl Climatol, 107(1–2), 265–277,
doi:10.1007/s00704-011-0464-2, 2012.
Jackson, L. S. and Forster, P. M.: An Empirical Study of Geographic and Seasonal Variations in
Diurnal Temperature Range, J. Climate, 23(12), 3205–3221, doi:10.1175/2010JCLI3215.1, 2010.
Jackson, T. J., Le Vine, D. M., Hsu, A. Y., Oldak, A., Starks, P. J., Swift, C. T., Isham, J. D. and
Haken, M.: Soil moisture mapping at regional scales using microwave radiometry: the Southern Great
Plains Hydrology Experiment, IEEE Transactions on Geoscience and Remote Sensing, 37(5), 2136–
2151, doi:10.1109/36.789610, 1999.
Jang, J.-D., Viau, A. A. and Anctil, F.: Neural network estimation of air temperatures from AVHRR
data, International Journal of Remote Sensing, 25(21), 4541–4554,
doi:10.1080/01431160310001657533, 2004.
Jarvis, P. G. and Mcnaughton, K. G.: Stomatal Control of Transpiration: Scaling Up from Leaf' to
Region, Advances in ecological research, 15, 49, 1986.
Juang, J.-Y., Katul, G., Siqueira, M., Stoy, P. and Novick, K.: Separating the effects of albedo from
eco-physiological changes on surface temperature along a successional chronosequence in the
southeastern United States, Geophysical Research Letters, 34(21), doi:10.1029/2007GL031296, 2007.
Jung, M., Koirala, S., Weber, U., Ichii, K., Gans, F., Camps-Valls, G., Papale, D., Schwalm, C.,
Tramontana, G. and Reichstein, M.: The FLUXCOM ensemble of global land-atmosphere energy
fluxes, Scientific Data, 6(1), doi:10.1038/s41597-019-0076-8, 2019.
Kilibarda, M., Hengl, T., Heuvelink, G. B. M., Gräler, B., Pebesma, E., Perčec Tadić, M. and Bajat, B.:
Spatio-temporal interpolation of daily temperatures for global land areas at 1 km resolution, J.
Geophys. Res. Atmos., 119(5), 2294–2313, doi:10.1002/2013JD020803, 2014.
Kröniger, K., De Roo, F., Brugger, P., Huq, S., Banerjee, T., Zinsser, J., Rotenberg, E., Yakir, D.,
Rohatyn, S. and Mauder, M.: Effect of Secondary Circulations on the Surface–Atmosphere Exchange
of Energy at an Isolated Semi-arid Forest, Boundary-Layer Meteorol, 169(2), 209–232,
doi:10.1007/s10546-018-0370-6, 2018.
Kustas, W. P. and Daughtry, C. S. T.: Estimation of the soil heat flux/net radiation ratio from spectral
data, Agricultural and Forest Meteorology, 49(3), 205–223, doi:10.1016/0168-1923(90)90033-3, 1990.





Kustas, W. P. and Norman, J. M.: Evaluation of soil and vegetation heat flux predictions using a simple
two-source model with radiometric temperatures for partial canopy cover, Agricultural and Forest
Meteorology, 94(1), 13–29, doi:10.1016/S0168-1923(99)00005-2, 1999.
Lee, X., Goulden, M. L., Hollinger, D. Y., Barr, A., Black, T. A., Bohrer, G., Bracho, R., Drake, B.,
Goldstein, A., Gu, L., Katul, G., Kolb, T., Law, B. E., Margolis, H., Meyers, T., Monson, R., Munger,
W., Oren, R., Paw U, K. T., Richardson, A. D., Schmid, H. P., Staebler, R., Wofsy, S. and Zhao, L.:
Observed increase in local cooling effect of deforestation at higher latitudes, Nature, 479(7373), 384–
387, doi:10.1038/nature10588, 2011.
Li, Y., Zhao, M., Motesharrei, S., Mu, Q., Kalnay, E. and Li, S.: Local cooling and warming effects of
forests based on satellite observations, Nat Commun, 6(1), 6603, doi:10.1038/ncomms7603, 2015.
Lindvall, J. and Svensson, G.: The diurnal temperature range in the CMIP5 models, Clim Dyn, 44(1–
2), 405–421, doi:10.1007/s00382-014-2144-2, 2015.
Liu, B., Xu, M., Henderson, M., Qi, Y. and Li, Y.: Taking China's Temperature: Daily Range,
Warming Trends, and Regional Variations, 1955–2000, J. Climate, 17(22), 4453–4462,
doi:10.1175/3230.1, 2004.
Loveland, T. R. and Belward, A. S.: The International Geosphere Biosphere Programme Data and
Information System global land cover data set (DISCover), Acta Astronautica, 41(4–10), 681–689,
doi:10.1016/S0094-5765(98)00050-2, 1997.
Luyssaert, S., Jammet, M., Stoy, P. C., Estel, S., Pongratz, J., Ceschia, E., Churkina, G., Don, A., Erb,
K., Ferlicoq, M., Gielen, B., Grünwald, T., Houghton, R. A., Klumpp, K., Knohl, A., Kolb, T.,
Kuemmerle, T., Laurila, T., Lohila, A., Loustau, D., McGrath, M. J., Meyfroidt, P., Moors, E. J.,
Naudts, K., Novick, K., Otto, J., Pilegaard, K., Pio, C. A., Rambal, S., Rebmann, C., Ryder, J., Suyker,
A. E., Varlagin, A., Wattenbach, M. and Dolman, A. J.: Land management and land-cover change have
impacts of similar magnitude on surface temperature, Nature Climate Change, 4(5), 389–393,
doi:10.1038/nclimate2196, 2014.
Makowski, K., Jaeger, E. B., Chiacchio, M., Wild, M., Ewen, T. and Ohmura, A.: On the relationship
between diurnal temperature range and surface solar radiation in Europe, J. Geophys. Res., 114,
D00D07, doi:10.1029/2008JD011104, 2009.
Mallick, K., Trebs, I., Boegh, E., Giustarini, L., Schlerf, M., Drewry, D. T., Hoffmann, L., von
Randow, C., Kruijt, B., Araùjo, A., Saleska, S., Ehleringer, J. R., Domingues, T. F., Ometto, J. P. H.
B., Nobre, A. D., de Moraes, O. L. L., Hayek, M., Munger, J. W. and Wofsy, S. C.: Canopy-scale
biophysical controls of transpiration and evaporation in the Amazon Basin, Hydrology and Earth
System Sciences, 20(10), 4237–4264, doi:10.5194/hess-20-4237-2016, 2016.
Mearns, L. O., Giorgi, F., McDaniel, L. and Shields, C.: Analysis of variability and diurnal range of
daily temperature in a nested regional climate model: comparison with observations and doubled CO2
results, Climate Dynamics, 11(4), 193–209, doi:10.1007/BF00215007, 1995.
Mildrexler, D. J., Zhao, M. and Running, S. W.: A global comparison between station air temperatures
and MODIS land surface temperatures reveals the cooling role of forests, Journal of Geophysical
Research, 116(G3), doi:10.1029/2010JG001486, 2011.
Oyler, J. W., Dobrowski, S. Z., Holden, Z. A. and Running, S. W.: Remotely Sensed Land Skin
Temperature as a Spatial Predictor of Air Temperature across the Conterminous United States, J. Appl.
Meteor. Climatol., 55(7), 1441–1457, doi:10.1175/JAMC-D-15-0276.1, 2016.
Panwar, A., Kleidon, A. and Renner, M.: Do Surface and Air Temperatures Contain Similar Imprints
of Evaporative Conditions?, Geophysical Research Letters, 46(7), 3802–3809,
doi:10.1029/2019GL082248, 2019.





Price, J. C.: Estimation of Regional Scale Evapotranspiration Through Analysis of Satellite Thermal-
infrared Data, IEEE Transactions on Geoscience and Remote Sensing, GE-20(3), 286–292,
doi:10.1109/TGRS.1982.350445, 1982.
Renner, M., Wild, M., Schwarz, M. and Kleidon, A.: Estimating Shortwave Clear-Sky Fluxes From
Hourly Global Radiation Records by Quantile Regression, Earth and Space Science, 6(8), 1532–1546,
doi:10.1029/2019EA000686, 2019.
Rotenberg, E. and Yakir, D.: Contribution of Semi-Arid Forests to the Climate System, Science,
327(5964), 451–454, doi:10.1126/science.1179998, 2010.
Saltzman, B. and Pollack, J. A.: Sensitivity of the Diurnal Surface Temperature Range to Changes in
Physical Parameters, Journal of Applied Meteorology, 16, 614–619, doi:https://doi.org/10.1175/1520-
0450(1977)016<0614:SOTDST>2.0.CO;2, 1977.
Shukla, J. and Mintz, Y.: Influence of Land-Surface Evapotranspiration on the Earth's Climate,
Science, 215(4539), 1498–1501, doi:10.1126/science.215.4539.1498, 1982.
Shuttleworth, W. J., Gurney, R. J., Hsu, A. Y. and Ormsby, J. P.: FIFE: the variation in energy partition
at surface flux sites., IAHS Publ 186, no. 6, 1989.
Stenchikov, G. L. and Robock, A.: Diurnal asymmetry of climatic response to increased $CO_2$ and
aerosols: Forcings and feedbacks, J. Geophys. Res., 100(D12), 26211, doi:10.1029/95JD02166, 1995.
Su, H., Wood, E. F., Mccabe, M. F. and Su, Z.: Evaluation of Remotely Sensed Evapotranspiration
Over the CEOP EOP-1 Reference Sites, Journal of the Meteorological Society of Japan, 85A, 439–459,
doi:10.2151/jmsj.85A.439, 2007.
Thom, A. S.: Momentum, mass and heat exchange of vegetation, Quarterly Journal of the Royal
Meteorological Society, 98(415), 124–134, doi:10.1002/qj.49709841510, 1972.
Thornthwaite, C. W.: An Approach toward a Rational Classification of Climate, Geographical Review,
38(1), 55, doi:10.2307/210739, 1948.
Tramontana, G., Jung, M., Camps-Valls, G., Ichii, K., Raduly, B., Reichstein, M., Schwalm, C. R.,
Arain, M. A., Cescatti, A., Kiely, G., Merbold, L., Serrano-Ortiz, P., Sickert, S., Wolf, S. and Papale,
D.: Predicting carbon dioxide and energy fluxes across global FLUXNET sites with regression
algorithms, , doi:10.5194/bg-2015-661, n.d.
Verma, S. B.: Aerodynamic resistances to transfers of heat, mass and momentum, Estimation of Areal
Evapotranspiration, T.A. Black, D.L. Spittlehouse, M.D. Novak and D.T. Price (ed)., IAHS, Pub. No.
177., 13–20, 1989.
Vinukollu, R. K., Wood, E. F., Ferguson, C. R. and Fisher, J. B.: Global estimates of
evapotranspiration for climate studies using multi-sensor remote sensing data: Evaluation of three
process-based approaches, Remote Sensing of Environment, 115(3), 801–823,
doi:10.1016/j.rse.2010.11.006, 2011.
Wang, K. and Dickinson, R. E.: Contribution of solar radiation to decadal temperature variability over
land, Proceedings of the National Academy of Sciences, 110(37), 14877–14882,
doi:10.1073/pnas.1311433110, 2013.
Wild, M.: From Dimming to Brightening: Decadal Changes in Solar Radiation at Earth's Surface,
Science, 308(5723), 847–850, doi:10.1126/science.1103215, 2005.
Yao, Y., Liang, S., Cheng, J., Liu, S., Fisher, J. B., Zhang, X., Jia, K., Zhao, X., Qin, Q., Zhao, B.,
Han, S., Zhou, G., Zhou, G., Li, Y. and Zhao, S.: MODIS-driven estimation of terrestrial latent heat
flux in China based on a modified Priestley–Taylor algorithm, Agricultural and Forest Meteorology,
171–172, 187–202, doi:10.1016/j.agrformet.2012.11.016, 2013.





Zhang, W., Huang, Y., Yu, Y. and Sun, W.: Empirical models for estimating daily maximum,
minimum and mean air temperatures with MODIS land surface temperatures, International Journal of
Remote Sensing, 32(24), 9415–9440, doi:10.1080/01431161.2011.560622, 2011.
Zhao, L., Lee, X., Smith, R. B. and Oleson, K.: Strong contributions of local background climate to
urban heat islands, Nature, 511(7508), 216–219, doi:10.1038/nature13462, 2014.
Zhu, W., Lü, A. and Jia, S.: Estimation of daily maximum and minimum air temperature using MODIS
land surface temperature products, Remote Sensing of Environment, 130, 62–73,
doi:10.1016/j.rse.2012.10.034, 2013.






























**Appendix**
**Table A1** Abbreviation used

| Symbol | Full form | Unit |
|---|---|---|
| DTR | Diurnal temperature range | K |
| $DT_sR$ | Diurnal surface temperature range | K |
| $R_s$ | Surface solar radiation | W m$^{-2}$ |
| $R_{s,max}$ | Maximum of surface solar radiation | W m$^{-2}$ |
| $T_a$ | 2 m air temperature | K |
| $T_s$ | Surface temperature, obtained from longwave radiation | K |
| $R_{l,net}$ | Net longwave radiation | W m$^{-2}$ |
| LE | Latent heat flux | W m$^{-2}$ |
| H | Sensible heat flux | W m$^{-2}$ |
| G | Ground heat flux | W m$^{-2}$ |
| $R_o$ | Net radiation at reference temperature | W m$^{-2}$ |
| $T_{ref}$ | Reference temperature | K |
| $k_r$ | Linearized constant | W m$^{-2}$ K$^{-1}$ |
| $\sigma$ | Stefan-Boltzmann constant | W m$^{-2}$ K$^{-4}$ |
| $c_p$ | Specific heat capacity of the lower atmosphere | J/kg K |
| $\rho$ | Density of the lower atmosphere | Kg m$^{-3}$ |
| $g_a$ | Aerodynamic conductance | m s$^{-1}$ |
| $u$ | Wind speed | m s$^{-1}$ |
| $u_*$ | Frictional velocity | m s$^{-1}$ |
| $f_e$ | Evaporative fraction | - |
| $\dfrac{dT_s}{dR_s}$ | Surface temperature warming rate | K/W m$^{-2}$ |
| $\dfrac{dT_a}{dR_s}$ | Air temperature warming rate | K/W m$^{-2}$ |
| $\dfrac{dT_s'}{dR_s}$ | Derivative of surface temperature warming rate to evaporative fraction | K/W m$^{-2}$ |
| $\dfrac{dT_a'}{dR_s}$ | Derivative of air temperature warming rate to evaporative fraction | K/W m$^{-2}$ |
| $g_a'$ | Derivative of aerodynamic conductance to evaporative fraction | m s$^{-1}$ |



**Table A2 Description** of sites used for this study

| Site no. | IGBP land use | Site ID | Site name | Location | | Number of days used | DOI |
|---|---|---|---|---|---|---|---|
| | | | | Latitude | Longitude | | |
| 1 | Croplands (CRO) | AU-Rig | Riggs Creek | -36.65 | 145.57 | 237 | https://doi.org/10.18140/FLX/1440202 |
| 2 | | CH-Oe1 | Oensingen1 grass | 47.28 | 7.73 | 182 | https://doi.org/10.18140/FLX/1440135 |
| 3 | | CZ-wet | CZECHWET | 49.02 | 14.77 | 184 | https://doi.org/10.18140/FLX/1440145 |
| 4 | | DE-Geb | Gebesee | 51.10 | 10.91 | 285 | https://doi.org/10.18140/FLX/1440146 |
| 5 | | IT-BCi | Borgo Cioffi | 40.52 | 14.95 | 274 | https://doi.org/10.18140/FLX/1440166 |





| | | | | | | |
|---|---|---|---|---|---|---|
| 6 | | IT-CA2 | Castel d`Asso2 | 42.37 | 12.02 | 143 | https://doi.org/10.18140/FLX/1440231 |
| 7 | | JP-SMF | Seto Mixed Forest Site | 35.25 | 137.06 | 164 | https://doi.org/10.18140/FLX/1440239 |
| 8 | | US-ARM | ARM Southern Great Plains site | 36.60 | -97.48 | 648 | https://doi.org/10.18140/FLX/1440066 |
| 9 | Croplands /Natural Vegetation (CRO/NV) | CH-Cha | Chamau grassland | 47.21 | 8.41 | 188 | https://doi.org/10.18140/FLX/1440131 |
| 10 | | CH-Fru | Fruebuel grassland | 47.11 | 8.53 | 260 | https://doi.org/10.18140/FLX/1440133 |
| 11 | | FR-LBr | Le Bray (after 6/28/1998) | 44.71 | -0.76 | 265 | https://doi.org/10.18140/FLX/1440163 |
| 12 | | US-Goo | 'Goodwin Creek' | 34.25 | -89.87 | 206 | https://doi.org/10.18140/FLX/1440070 |
| 13 | Grasslands (GRA) | AU-Stp | Sturt Plains | -17.15 | 133.35 | 532 | https://doi.org/10.18140/FLX/1440204 |
| 14 | | IT-MBo | Monte Bondone | 46.01 | 11.04 | 480 | https://doi.org/10.18140/FLX/1440170 |
| 15 | | US-AR1 | ARM USDA UNL OSU Woodward Switchgrass 1 | 36.42 | -99.42 | 242 | https://doi.org/10.18140/FLX/1440103 |
| 16 | | US-AR2 | ARM USDA UNL OSU Woodward Switchgrass 2 | 36.63 | -99.59 | 225 | https://doi.org/10.18140/FLX/1440104 |
| 17 | | US-SRG | Santa Rita Grassland | 31.78 | -110.82 | 696 | https://doi.org/10.18140/FLX/1440114 |
| 18 | | US-Wkg | Walnut Gulch Kendall Grasslands | 31.73 | -109.94 | 1074 | https://doi.org/10.18140/FLX/1440097 |
| 19 | Shrublands (SH) | AU-ASM | Alice Springs | -22.28 | 133.24 | 477 | https://doi.org/10.18140/FLX/1440194 |
| 20 | | US-SRC | Santa Rita Creosote | 31.90 | -110.83 | 621 | https://doi.org/10.18140/FLX/1440098 |
| 21 | | US-SRM | Santa Rita Mesquite | 31.82 | -110.86 | 1121 | https://doi.org/10.18140/FLX/1440090 |
| 22 | | US-Whs | Walnut Gulch Lucky Hills Shrubland | 31.74 | -110.05 | 558 | https://doi.org/10.18140/FLX/1440097 |
| 23 | Savannas (SA) | AU-Cpr | Calperum | -34.00 | 140.58 | 284 | https://doi.org/10.18140/FLX/1440195 |
| 24 | | AU-DaP | Daly River Pasture | -14.06 | 131.31 | 439 | https://doi.org/10.18140/FLX/1440123 |
| 25 | | AU-DaS | Daly River Savanna | -14.15 | 131.38 | 504 | https://doi.org/10.18140/FLX/1440122 |
| 26 | | AU-Dry | Dry River | -15.25 | 132.37 | 466 | https://doi.org/10.18140/FLX/1440197 |
| 27 | | AU-How | Howard Springs | -12.49 | 131.15 | 355 | https://doi.org/10.18140/FLX/1440125 |
| 28 | Woody Savannas (WSA) | AU-Gin | Gingin | -31.37 | 115.65 | 212 | https://doi.org/10.18140/FLX/1440199 |
| 29 | | AU-Whr | Whroo | -36.67 | 145.02 | 206 | https://doi.org/10.18140/FLX/1440206 |
| 30 | | IT-Noe | Sardinia/Arca di Noe | 40.60 | 8.15 | 555 | https://doi.org/10.18140/FLX/1440171 |
| 31 | | US-Me6 | Metolius New Young Pine | 44.32 | -121.60 | 270 | https://doi.org/10.18140/FLX/1440099 |
| 32 | | US-Var | Vaira Ranch | 38.40 | -120.95 | 1091 | https://doi.org/10.18140/FLX/1440094 |
| 33 | Deciduous Broadleaf Forest (DBF) | DK-Sor | Soroe- LilleBogeskov | 55.48 | 11.64 | 169 | https://doi.org/10.18140/FLX/1440155 |
| 34 | | IT-Col | Collelongo- Selva Piana | 41.84 | 13.58 | 343 | https://doi.org/10.18140/FLX/1440167 |
| 35 | | US-Oho | Oak Openings | 41.55 | -83.84 | 408 | https://doi.org/10.18140/FLX/1440088 |
| 36 | | US-WCr | Willow Creek | 45.80 | -90.07 | 237 | https://doi.org/10.18140/FLX/1440095 |
| 37 | Evergreen Broadleaf Forest | AU-Wom | Wombat | -37.42 | 144.09 | 180 | https://doi.org/10.18140/FLX/1440207 |
| 38 | | CA-Obs | SK-Southern Old Black Spruce | 53.98 | -105.11 | 620 | https://doi.org/10.18140/FLX/1440044 |
| 39 | | CA-Qfo | Quebec Eastern Old Black Spruce (EOBS) | 49.69 | -74.34 | 194 | https://doi.org/10.18140/FLX/1440045 |
| 40 | | DE-Tha | Tharandt- Anchor Station | 50.96 | 13.56 | 268 | https://doi.org/10.18140/FLX/1440152 |



| 41 | Evergreen Needleleaf Forest | IT-Lav | Lavarone (after 3/2002) | 45.95 | 11.28 | 557 | https://doi.org/10.18140/FLX/1440169 |
|----|------|--------|-------------------------|-------|--------|-----|--------------------------------------|
| 42 | (ENF) | IT-Ren | Renon/Ritten (Bolzano) | 46.58 | 11.43 | 362 | https://doi.org/10.18140/FLX/1440173 |
| 43 | | NL-Loo | Loobos | 52.16 | 5.74 | 401 | https://doi.org/10.18140/FLX/1440178 |
| 44 | | US-GLE | GLEES | 41.36 | -106.23 | 514 | https://doi.org/10.18140/FLX/1440069 |
| 45 | | US-Me2 | Metolius Intermediate Pine | 44.45 | -121.55 | 450 | https://doi.org/10.18140/FLX/1440079 |
| 46 | | US-NR1 | Niwot Ridge (LTER NWT1) | 40.03 | -105.54 | 600 | https://doi.org/10.18140/FLX/1440087 |
| 47 | | CA-Gro | ON-Groundhog River Mixedwood | 48.21 | -82.15 | 339 | https://doi.org/10.18140/FLX/1440034 |
| 48 | Mixed Forest | CA-Oas | SK-Old Aspen | 53.62 | -106.19 | 688 | https://doi.org/10.18140/FLX/1440043 |
| 49 | (MF) | CA-TP4 | ON-Turkey Point 1939 White Pine | 42.70 | -80.35 | 482 | https://doi.org/10.18140/FLX/1440053 |
| 50 | | FR-Pue | Puechabon | 43.74 | 3.59 | 535 | https://doi.org/10.18140/FLX/1440164 |
| 51 | | RU-Fyo | Fedorovskoje-drained spruce stand | 56.46 | 32.92 | 257 | https://doi.org/10.18140/FLX/1440183 |




**Observational analysis for each site**

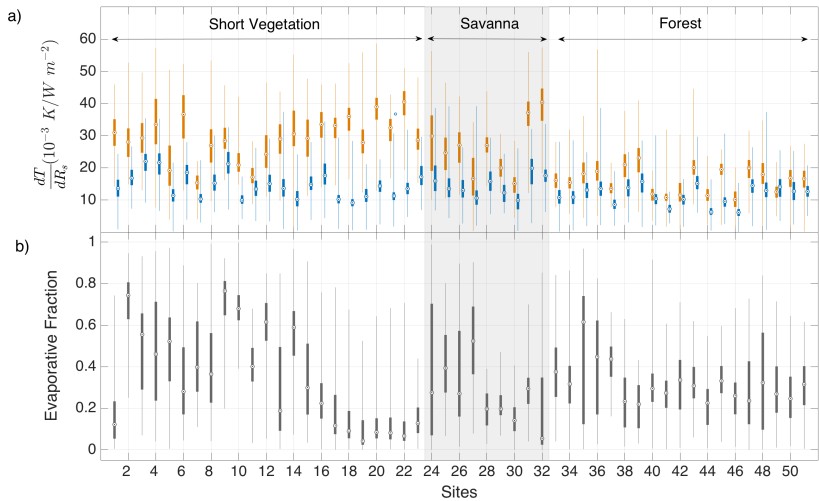


**Figure A1** (a) Box plot of surface ($T_s$, orange) and air ($T_a$, blue) temperature warming rates ($dT/dR_s$),
(b) Box plot of evaporative fractions. The vegetation types are separated by grey and white shades. The
circle in the box plot indicates the median and the top and bottom edges indicate the 75[th] and 25[th]
percentiles, respectively. The whisker covers the range in the observation.




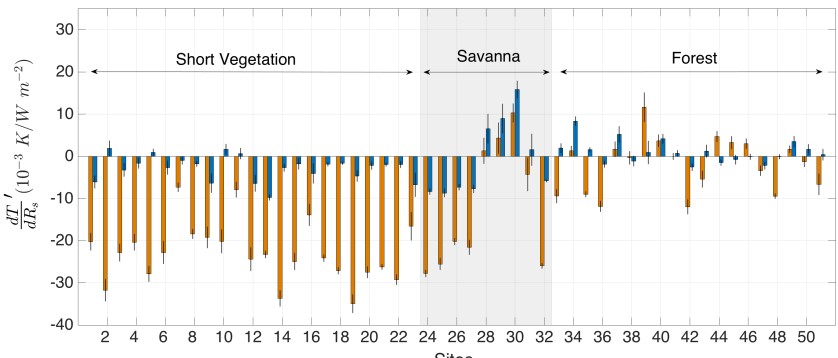

**Figure A2** Warming rate response to evaporation ($dT'/dR_s$) for surface ($T_s$, orange) and air ($T_a$, blue) temperature. The vegetation types are separated by grey and white shades. The black bar represents the standard error in the linear regression of observed warming rate and evaporative fraction.

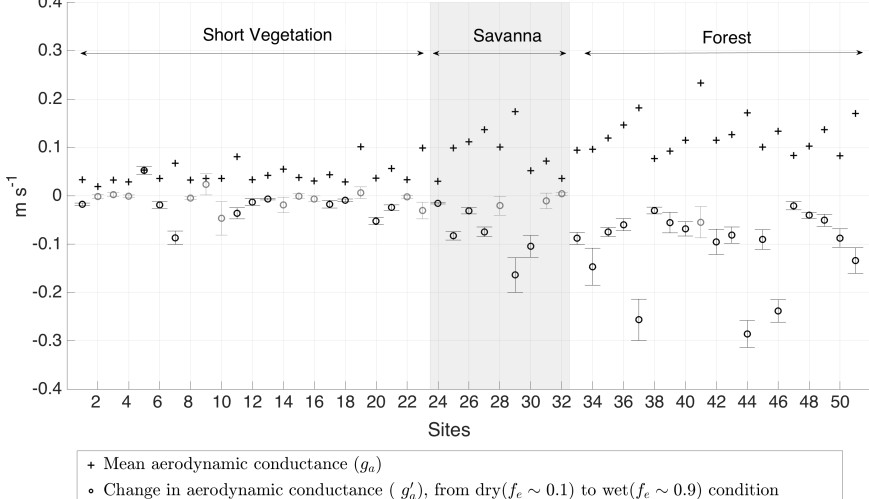

**Figure A3** Mean aerodynamic conductance ($g_a$) and response of aerodynamic conductance to evaporation ($g_a'$) for each site. Sites with non-significant $g_a'$ in observation is marked by light shades. The vegetation types are separated by grey and white shades. The error bar represents the standard error in the observed linear regression of aerodynamic conductance and evaporative fraction.