# Peer review of "Imprints of evaporative conditions and vegetation type in"

_Hydrology and Earth System Sciences, 2020_

## Referee Comment (RC1) · Anonymous Referee #1 · 1 Apr 2020

The study by Panwar proposed a metric "warming rate" to quantify the diurnal variation in surface and air temperature, and investigated the sensitivity of the warming rate to evaporative fraction and aerodynamic conductance in different land cover conditions. The authors concluded that the surface warming rate is sensitivity to the evaporative fraction over the short vegetation, but no imprint of evaporation is found in the diurnal surface/air temperature. This is an interesting study, and it is on a topic of relevance and general interest to the readers of this journal. However, some parts of the manuscript are hard to follow, and need some improvements and better discussions.

My major concerns are:

1. What is the purpose of calculating the temperature warming rate to evaporative fraction? Previous studies suggest that aerodynamic resistance (or conductance) plays

a major role in land use-induced temperature change (Zhao et al. 2014; Chen and Dirmeyer 2016; Winckler et al. 2019). The surface warming rate can be mainly associated with aerodynamic conductance. The consistency between the distribution of the warming rate in Figure 3 and the distribution of aerodynamic conductance in Figure 5 may support this idea - higher conductance corresponds to a lower warming rate. How is the distribution of the evaporative fraction? And why not calculate the sensitivity of "the sensitivity" to aerodynamic resistance, but only use aerodynamic conductance as a secondary factor?

2. One of the main conclusions of this study is that the warming rates do not carry imprints of evaporation in the forest. What is the physical mechanism of this absent imprints? Why does the high aerodynamic conductance of the forest result in no imprint of evaporation in diurnal temperature variations? Does this necessarily mean ET of the forest has little impact on temperature? The authors need to provide more explanations of the physical processes and more discussions on the implications of the warming rate sensitivity.

3. Throughout the manuscript, "evaporation (or ET)" and "evaporative fraction" are used as interchangeable terms. However, the evaporative fraction also carries information about the sensible heat flux. I am wondering what the sensitivity of the warming rate to ET based on the observational analysis.

4. A similar issue to my 3rd point - the authors should note that aerodynamic conductance and evaporative fraction are not independent (Rigden and Li 2017). However, in their analysis, aerodynamic conductance and evaporative fraction were treated as two independent factors that govern the diurnal variations of surface temperature. For instance, "model reveals a strong sensitivity of the warming rates to evaporative fraction and aerodynamic conductance"; "the diurnal variations in temperatures are mainly governed by their aerodynamic properties resulting in no imprint of evaporation in diurnal temperature variations." If considering the dependency of aerodynamic conductance and evaporative fraction, will their conclusions be the same?

Specific comments:

1. L28-29: Why use diurnal temperature variations to predict evaporation? And this conclusion seems opposite to the conclusion in L567-568.

2. L95: Define surface temperature. Skin temperature? I noted it is explained in the later section, but it would be better to explain it here.

3. Section 2 should be a subsection of section 3 or at least after section 3. It should be a consistent order with the structure of the result section.

4. L166: How to get the reference temperature Tref?

5. L187-188: Which two equations?

6. L208-211: Why are these two filters used? Can the authors provide more explanations for the first filter?

7. L220: The FLUXNET site also provides information about the vegetation type. Is the reported land cover type consistent with the 1km IGBP land cover product?

8. L277-278: The difference should be attributed to the low aerodynamic conductance? For the short vegetation, the conductance is low, which means the heat transfer from the surface to the lower atmosphere is less efficient.

9. L292-293: Where is the consistency in Figure A2?

10. L293-295: It is very confusing here. First, this sentence does not explain anything from the previous sentence. Second, Figure 4 only shows the sensitivity of the warming rate. Is the actual range of the evaporative fraction 0~1 from dry to wet days? In methodology (L167), the range is defined as 0.1~0.9?

11. Figure 4: The label of the color bar is not correct.

12. L369: Figure 6a?

13. L373: same issue as above.

14. L390-391: What does each dot stand for in Figure 6b?

15. L406-407: "the response of the surface temperature warming rate to evaporative fraction is predominantly governed by the aerodynamic conductance" - Can you explain how to get this conclusion from Figure 6?

16. L436-437: "enhanced aerodynamic conductance plays a small, but noticeable role in weakening the response of the warming rate to evaporative fraction." - This conclusion is contradictory to the conclusion in L406-407.

Reference:

Chen, L., & Dirmeyer, P. A. (2016). Adapting observationally based metrics of biogeophysical feedbacks from land cover/land use change to climate modeling. Environmental Research Letters, 11(3), 034002.

Rigden, A. J., & Li, D. (2017). Attribution of surface temperature anomalies induced by land use and land cover changes. Geophysical Research Letters, 44(13), 6814-6822.

Winckler, J., Reick, C. H., Bright, R. M., & Pongratz, J. (2019). Importance of surface roughness for the local biogeophysical effects of deforestation. Journal of Geophysical Research: Atmospheres, 124(15), 8605-8618.

Zhao, L., Lee, X., Smith, R. B., & Oleson, K. (2014). Strong contributions of local background climate to urban heat islands. Nature, 511(7508), 216-219.

---

## Referee Comment (RC2) · Anonymous Referee #2 · 11 May 2020

This paper investigates the response of the diurnal warming rate of the surface and air temperature to evaporative conditions and vegetation cover type, which could be useful, as the authors point out, when estimating air temperatures from remote sensing of surface temperatures. They develop a simple model for the warming rate based on the surface energy balance which captures its observed response to ga and fe reasonably well.

Overall, the idea is good and the study is thorough, so I recommend publication after revision of some issues. Some of these issues have already been addressed by my fellow referee.

My main difficulty is to see why, when deriving equation (4), both the evaporative fraction and the aerodynamic conductance can be considered constant with Rs. This

needs some justification. Later in the paper it is mentioned (L241-242) that evaporative fraction is stable during daylight hours, which should probably mentioned before presenting Eq. (4). Why can the diurnal variation of ga be ignored?

The paper would in general benefit from a language revision.

Some minor comments:

- Plots using green and red are going to be difficult to read for colorblind people.

- The dashed lines in Fig. 1 are not very easy to see

- In Fig. 5, may it be more useful to express the ga' in the inset plot in relative terms (e.g. as a percentage of the mean aerodynamic conductance for each vegetation type)?

- L369 - Figure 6a

- L406 - Figure 6 (a or b?)

- L445 - Where does this 74% come from?

Some language and typos:

- L73 "... the warming rate, (comma) that eliminates..."

- L225 "Vegetation are classified into three types that is based on...". Rather "Vegetation IS classified into three types, based on..."?

- L251 - The year in bracketed citations shouldn't be between brackets itself (see, e.g., Verma, 1989)

- L427 - ".. contribution of the contribution..."

- L487 - depend/depends

- L533 - ambiguity? (uncertainty?)
* * *
95, 2020.

---

## Author Comment (AC1) · 13 May 2020

**Response to the reviewer # 1 HESSD**

Dear Reviewer,

Thank you. We appreciate the time and effort that you have dedicated to providing your valuable feedback on our manuscript. We found your comments extremely helpful and we have been able to incorporate changes to reflect most of the suggestions. These changes are highlighted in "*italic*" with the line numbers in this response letter, and in red color within the manuscript. The line numbers might change with future edition of the manuscript.

Your comments are in blue bold color, which we answer in the black color. Some of the compre-

hensive questions are split within the comment in blue font. Figure Rn denotes figures which are in the supplement of this response; here n is the figure number.
**on** "**Imprints of evaporation and vegetation type in diurnal temperature variations**" **by Annu Panwar et al.**

**Anonymous Referee # 1**

**The study by Panwar proposed a metric** "**warming rate**" **to quantify the diurnal variation in surface and air temperature, and investigated the sensitivity of the warming rate to evaporative fraction and aerodynamic conductance in different land cover conditions. The authors concluded that the surface warming rate is sensitivity to the evaporative fraction over the short vegetation, but no imprint of evaporation is found in the diurnal surface/air temperature. This is an interesting study, and it is on a topic of relevance and general interest to the readers of this journal. However, some parts of the manuscript are hard to follow, and need some improvements and better discussions.**

We are glad that you find this study interesting and relevant to HESS journal. To further improve our manuscript we have gone through each of your comments.

**My major concerns are:**

1. **What is the purpose of calculating the temperature warming rate to evaporative fraction?**

    **Previous studies suggest that aerodynamic resistance (or conductance) plays a major role in land use-induced temperature change (Zhao et al. 2014; Chen and**

[Figure]

**Dirmeyer 2016; Winckler et al. 2019). The surface warming rate can be mainly asso-
ciated with aerodynamic conductance. The consistency between the distribution
of the warming rate in Figure 3 and the distribution of aerodynamic conductance
in Figure 5 may support this idea - higher conductance corresponds to a lower
warming rate. How is the distribution of the evaporative fraction? And why not
calculate the sensitivity of "the sensitivity" to aerodynamic resistance, but only
use aerodynamic conductance as a secondary factor?**

We assume the reviewer wants to ask "what is the purpose of calculating the response
of temperature warming rate to evaporative fraction?"

In this study we show if diurnal temperature responds to evaporative conditions in differ-
ent vegetation types. For this, we use warming rate because it removes the dominant
contribution of the solar radiation and allows us to investigate the effects of evaporative
conditions and vegetation types in diurnal temperature variation. Expression of warming
rate is also directly related to evaporative fraction and aerodynamic conductance of veg-
etation, which is obtainable from surface energy balance. Evaporation cools that means
any heating of diurnal temperature due to unit absorption of solar radiation will dampen
due to evaporative cooling. To quantify this effect we look at the relationship of warming
rates to evaporative fraction.

Previous studies suggest that aerodynamic resistance (or conductance) plays a major
role in land use-induced temperature change (Zhao et al. 2014; Chen and Dirmeyer
2016; Winckler et al. 2019).

We completely agree that the aerodynamic conductance plays a major role in land use-
induced temperature change. Our findings also show that the warming rates of forests
are lower predominantly due to their high aerodynamic conductance (L415-416).

We are aware of the listed studies that capture the impact of vegetation properties in
temperature, in fact we have already cited them (Juang et al., 2007; Lee et al., 2011;

Luyssaert et al., 2014; Zhao et al., 2014) in the introduction section, see L67-70.

These studies are relevant to our work but not the same. The cited studies show the change in surface energy balance components due to land cover change, which leads the changes in temperature. On the other hand our study looks at the diurnal scale and explore the local impact of variation in evaporative conditions and aerodynamic conductance on diurnal course of surface and air temperature in different vegetation types.

The surface warming rate can be mainly associated with aerodynamic conductance. The consistency between the distribution of the warming rate in Figure 3 and the distribution of aerodynamic conductance in Figure 5 may support this idea - higher conductance corresponds to a lower warming rate.

We partially agree with this notion. However, surface temperature warming rate are mainly associated to the aerodynamic conductance only in forests but not in the short vegetation. Eqn. 5 in the manuscript shows, the variation in warming rate is not just dependent on aerodynamic conductance of vegetation but also on the evaporative conditions. As result, surface temperature warming rate of the short vegetation are higher on dry days and lower on the wet days (evaporative days), Figure 6a. This sensitivity is also captured in the observations, Figure 4.

How is the distribution of the evaporative fraction?

The range in evaporative fraction for each site is already present in the manuscript; see Figure A1.b in the Appendix. Additionally, the density distribution of the evaporative fraction is provided in the supplement of this response, Figure R1.

And why not calculate the sensitivity of "the sensitivity" to aerodynamic resistance, but

only use aerodynamic conductance as a secondary factor?

This is exactly what we show. Our model demonstrates that (Eq. 6) the warming rates are not only sensitive to aerodynamic conductance but also to the sensitivity of aerodynamic conductance to evaporative fraction. Figure 5 shows that the aerodynamic conductance increases on dry days, which we call as enhanced aerodynamic conductance . The sensitivity of warming rates to both the factors is quantified and compared in Figure 7b. Our study found that the aerodynamic conductance is the primary factor but the enhanced aerodynamic conductance is also somewhat important especially in the forests, L438

2. **One of the main conclusions of this study is that the warming rates do not carry imprints of evaporation in the forest. What is the physical mechanism of this absent imprints? Why does the high aerodynamic conductance of the forest result in no imprint of evaporation in diurnal temperature variations? Does this necessarily mean ET of the forest has little impact on temperature? The authors need to provide more explanations of the physical processes and more discussions on the implications of the warming rate sensitivity.**

We have split the response into several parts. Additional explanations of the physical mechanism and implication of warming rate is also added in the discussion section.

What is the physical mechanism of this absent imprints?

The physical mechanism explaining the insensitivity of diurnal temperatures to evaporative conditions in the forests is still unclear. It is not the main objective of the study but certainly one of the next questions to answer.

So far, our model explains the absent imprints of evaporation in forest by their high aerodynamic conductance. Also, the aerodynamic conductance of forest enhances on dry days, which means enhanced buoyancy in forest on dry days. Based on our results and

model we speculate that the enhanced buoyancy on dry days might explain the reduced warming rates and their weak response to evaporative fraction in the forests. This is now expressed in the discussion section, L537-542.

*"Our model shows, diurnal temperature variation in forests has weak to no imprints of evaporative conditions due to their high aerodynamic conductance. Additionally, observations show enhancement of aerodynamic conductance of forest on dry days, which also contributes in reduced warming of surface temperature. Based on our findings, we speculate that the enhanced aerodynamic conductance increases the buoyancy that may explain the physical mechanism causing the lower warming rates and their weaker response to evaporative conditions in the forest."*

Why does the high aerodynamic conductance of the forest result in no imprint of evaporation in diurnal temperature variations?

This is explained in Eq. 5 of the model, which demonstrates the dependency of surface temperature warming rate to evaporative fraction and aerodynamic conductance.

$$\frac{dT_s}{dR_s} \approx \frac{(1-f_e)}{c_p \; \rho g_a} + \frac{dT_a}{dR_s} \qquad\qquad Eq.\,(5)$$

In the above equation the denominator term ( $c_p \; \rho g_a$ is higher for the forest because of their high aerodynamic conductance than the numerator term $(1-f_e)$ . That means Eq. 5 is not very sensitive to variations in evaporative fraction. This sensitivity is also demonstrated in Figure 6a of the manuscript.

Does this necessarily mean ET of the forest has little impact on temperature?
[Figure]

ET or evaporative fraction of the forest has little impact on the diurnal variation of the temperature. It does not necessarily mean that the mean temperatures of the forests are insensitive to evaporation.

To indicate this in the text, we have now added the following line in the discussion, L532-535

"*It is also reflected in similar diurnal variations of surface and air temperature that are mostly insensitive to changes in evaporative fraction. Thus, evaporative fraction has little to no impact on the diurnal variation of temperatures in the forest. But, it does not necessarily indicate that the mean temperatures of forest are insensitive to evaporation.*"

3. **Throughout the manuscript, "evaporation (or ET)" and "evaporative fraction" are used as interchangeable terms. However, the evaporative fraction also carries information about the sensible heat flux. I am wondering what the sensitivity of the warming rate to ET based on the observational analysis.**

Thank you for pointing it out. In order to avoid such confusion we have replaced "evaporation" with "evaporative fraction" . It also helps in simplifying the model description.

This change is implemented throughout the text. This also applies to the title, which now reads as:

" *Imprints of evaporative condition and vegetation type in diurnal temperature variations*"

However, the evaporative fraction also carries information about the sensible heat flux.

We agree that evaporative fraction carries information about the sensible heat flux. Besides latent heat flux, sensible heat flux also cools the surface temperature, which is depicted in Eq. 3. Therefore, it is important to account all the turbulent heat fluxes to determine surface temperature.

Moreover, in the text we refer evaporative conditions as dry or less evaporative, when turbulent heat flux is dominated by sensible heat flux, and wet or evaporative, when turbulent heat flux is dominated by latent heat flux. This notion is now updated in the text (L104-106).

"*We observed that the warming rate of surface temperature decreases from dry (less-evaporative, sensible heat flux dominates) to wet (evaporative, latent heat flux dominates) conditions but the warming rate of air temperature remained unaffected by evaporative conditions.*"

I am wondering what the sensitivity of the warming rate to ET based on the observational analysis.

As depicted in Eq. 3, warming rate of surface temperature is not only dependent on LE or ET but also on sensible heat flux (H).

$$\frac{dT_s}{dR_s} = \frac{1}{k_r} - \frac{1}{k_r} \frac{d(H+LE)}{dR_s} \qquad\qquad Eq.\,(3)$$

We opt evaporative fraction because it contains information on both the turbulent heat fluxes, LE and H. Only taking LE is not numerically correct as per our model. Although, one can still see how warming rate responds to LE in the observations, Figure R2 in the supplement of this response.

4. **A similar issue to my 3rd point - the authors should note that aerodynamic conductance and evaporative fraction are not independent (Rigden and Li 2017). However, in their analysis, aerodynamic conductance and evaporative fraction were treated as two independent factors that govern the diurnal variations of surface temperature. For instance,** "**model reveals a strong sensitivity of the warming rates to evaporative fraction and aerodynamic conductance**" ; "**the diurnal variations in temperatures are mainly governed by their aerodynamic properties resulting in no imprint of evaporation in diurnal temperature variations.**" **If considering the dependency of aerodynamic conductance and evaporative fraction, will their conclusions be the same?**

We agree that aerodynamic conductance and evaporative fraction are not independent. To account for this, we also show the observed sensitivity of aerodynamic conductance to evaporative fraction and its consequences on warming rate. Your concern that this sensitivity should also reflect in warming rate is valid and already accounted in our study.

Eq. (5) calculates the warming rate using evaporative fraction ( $f_e$ and aerodynamic conductance ( $g_a$ .

$$\frac{dT_s}{dR_s} \approx \frac{(1 - f_e)}{c_p \, \rho g_a} + \frac{dT_a}{dR_s} \qquad\qquad Eq. \, (5)$$

We use daily observations of evaporative fraction and aerodynamic conductance to calculate daily surface temperature warming rates (Figure 6b). That means, $g_a$ is not the mean of the site and varies with $f_e$ . So, yes we do not consider a constant aerodynamic conductance and it is sensitive to $f_e$ . To review this concern in the manuscript we have added following sentences in the model and result section.

L180-184

"*Eq. (5) shows that morning to noon warming of surface temperature is a function of evaporative fraction, aerodynamic conductance and also of the warming rate of air temperature. Generally, vegetation has a characteristic aerodynamic conductance that depends on the surface roughness but it is important to consider that their daily aerodynamic conductance can vary with the daily evaporative conditions.*"

L405-406

"*Also, daily $g_a$ is not independent of the daily evaporative conditions, so their dependency is certainly captured here.*"

"the diurnal variations in temperatures are mainly governed by their aerodynamic properties resulting in no imprint of evaporation in diurnal temperature variations." If considering the dependency of aerodynamic conductance and evaporative fraction, will their conclusions be the same?

Yes we have already considered the dependency of aerodynamic conductance to evaporative fraction. Our conclusion is based on Eq. (6), where $g_a^{'}$ captures the sensitivity of $g_a$ to $f_e$ . Generally, $g_a$ is observed to increase on dry days, which we also call as enhanced aerodynamic conductance.

$$\frac{dT_s}{dR_s}^{'} = -\frac{1}{c_p \cdot \rho . g_a} - \frac{1 - f_e}{c_p \cdot \rho . g_a} \cdot \frac{g_a^{'}}{g_a} + \frac{dT_a}{dR_s}^{'} \qquad Eq.\,(6)$$

Figure 7b demonstrates that the sensitivities of warming rate to evaporative fraction will be slightly different if you consider $g_a^{'} \neq 0$ than the case when $g_a^{'} = 0$ . In forest enhanced aerodynamic conductance are important than in the short vegetation. But overall it is the high aerodynamic conductance of forest that reduces the responses of warming rate to evaporative fraction.

**Specific comments:**

1. **L28-29: Why use diurnal temperature variations to predict evaporation? And this conclusion seems opposite to the conclusion in L567-568.**

   The conclusions in the abstract (L28-29) and in the conclusion (L567-568) sections are exactly the same. See the sentence in L581-582

   "*This implies that the diurnal variation of surface temperature of short vegetation should be useful for estimating evaporation.*"

   Why use diurnal temperature variations to predict evaporation?

   We use diurnal temperature because it is one of the well-observed variables in meteorology. In a diurnal scale solar radiation directly warms the temperature but one may speculate that the enhanced evaporation will slow down this warming.

   We found that warming rate of surface temperature decreases linearly with evaporative fraction. This relationship can be useful in estimating evaporation, which is one of the important but rarely measured variable in meteorology.

   And this conclusion seems opposite to the conclusion in L567-568.

   No, they are not opposite.

   L567-568, "… *diurnal temperature variations can be understood and predicted by relatively few factors, solar radiation, aerodynamic conductance and evaporative fraction.*" summarizes the last figure (figure 8), where we show that the diurnal temperature range

can be estimated from solar radiation, aerodynamic conductance and evaporative fraction using the model. It is a general statement irrespective of the vegetation type. However, the importance of these three factors varies with vegetation. For example information of evaporative fraction is not necessarily important for forests but very important for short vegetation, which is explained in lines L74-475 and L479-481.

2. **L95: Define surface temperature. Skin temperature? I noted it is explained in the later section, but it would be better to explain it here.**

To resolve it, we have added "or skin" in bracket, in L96.

"*Figure 1a shows a greater surface (or skin) temperature warming rate compared to air temperature warming rate for a cropland site.*"

3. **Section 2 should be a subsection of section 3 or at least after section 3. It should be a consistent order with the structure of the result section.**

The model section is presented before the data and method section, because we think it is convenient for the reader to first understand what data one needs in order to conduct this analysis based on our model. To illustrate this in the text we have added the following sentence in the introduction, L125-127

"*Based on the model formulation we identify the data needed for the study. Description of data and methodology is given in data and method section.*"

We have also added an additional sentence supporting why we introduce model in section 2, L138-140

"*To understand how diurnal temperature variation responds to evaporative condition and*

*vegetation types, we first present quantitative expressions of diurnal temperature warming rate based on surface energy balance.*"

4. **L166: How to get the reference temperature Tref?**

The $T_{ref}$ in Eq.2 is the global mean of surface temperature which is $\sim$ 288 K, L166. This information is also added in the text, L175-177.

"*To get $k_r$ we consider $T_{ref} \sim$ 288 K which is the mean global surface temperature such that the term $k_r$ $(1 - f_e)$ varies by $\sim$ 4.87 Wm$^{-2}$ K$^{-1}$ to $\sim$ 0.54 Wm$^{-2}$ K$^{-1}$ from dry ( $f_e$ =0) to wet ( $f_e$ =1) conditions*
"

Nevertheless, choice of $T_{ref}$ does not affect the model outputs. The model is relatively insensitive to any particular values of $T_{ref}$ because the effect of term $k_r$ $(1 - f_e)$ is ignored in the model due to its smaller magnitude.

5. **L187-188: Which two equations?**

Corrected, see line L197-198

"*On multiplying Eq. (5) with daily maximum solar. . .*"

6. **L208-211: Why are these two filters used? Can the authors provide more explanations for the first filter?**

Author realizes that the sentence at L208-209 complicates the understanding of the filters used for data analysis, so it has been removed.

The first filter is used to select summer days (L202-205) and the second filter selects the clear sky days among the summer days (L209-211).

Can the authors provide more explanations for the first filter?

As explained in L209-211 the days with solar radiation greater than the median of its annual distribution are treated as summer days. Figure R3 in the supplement of this response demonstrates our methodology:

7. **L220: The FLUXNET site also provides information about the vegetation type. Is the reported land cover type consistent with the 1km IGBP land cover product?**

Yes, we used the land cover information provided by FLUXNET website which can be found here:

https://daac.ornl.gov/FLUXNET/guides/Fluxnet_site_DB.html# datasetoverview

A reference to this dataset is now added, L24-225.

"*The information on the vegetation type is obtained from the FLUXNET land cover classification (Falge et al., 2017) that is based on the International Geosphere-Biosphere Programme (IGBP) Data and Information System*"

8. **L277-278: The difference should be attributed to the low aerodynamic conductance? For the short vegetation, the conductance is low, which means the heat**

**transfer from the surface to the lower atmosphere is less efficient.**

We completely agree that the major differences in the surface temperature warming rates should be attributed to aerodynamic conductance. Lines L277-278 discuss the possible reasons for the spread in surface temperature warming rate within the short vegetation type.

There was a typing error; it was not "*air temperature warming rate*" but "*surface temperature warming rate of cropland sites*". See the corrected sentence below L279-281

"*Within the short vegetation type, grassland and shrubland sites show much greater surface temperature warming rates than the surface temperature warming rate of the cropland sites*"

9. **L292-293: Where is the consistency in Figure A2?**

Figure A2 shows the warming rate responses to evaporative fraction for each site. The word ???consistency??? was used for ???similar??? negative responses in all the sites of short vegetation type.

To avoid such confusion we have rephrased the sentence by replacing ???consistency??? term with the term ???similar???

See the modified version, L300-303

"*Short vegetation shows a mean decrease of $\sim$ 23 x 10$^{-3}$ K/W m$^{-2}$ in surface temperature warming rate from dry to wet condition. However, air temperature warming rate decreases only by $\sim$ 5 x 10$^{-3}$ K/W m$^{-2}$. Similar responses were reported in our previous study for a cropland site (Figure A2, site no. 8).*"

10. **L293-295: It is very confusing here. First, this sentence does not explain anything from the previous sentence. Second, Figure 4 only shows the sensitivity of the warming rate. Is the actual range of the evaporative fraction 0_1 from dry to wet days? In methodology (L167), the range is defined as 0.1_0.9?**

L293-295: It is very confusing here. First, this sentence does not explain anything from the previous sentence. Second, Figure 4 only shows the sensitivity of the warming rate.

Description of appendix figures is now removed in the main manuscript in order to avoid any confusion with Figure 4.

Is the actual range of the evaporative fraction 0_1 from dry to wet days? In methodology (L167), the range is defined as 0.1_0.9?

Mathematically a unit change in evaporative fraction is 0 to 1 but physically this range is unreasonable for vegetated surface. Therefore, to obtain the response of warming rate to unit change in evaporative fraction we use the slope of linear regressions of observed warming rate and evaporative fraction.

Assumption that evaporative fraction is $\sim 0.1$ on dry days and $\sim 0.9$ on wet days in the model section is discard by sticking to the standard mathematical definition of unit chance in evaporative fraction. Such that L167 reads:

"*Considering $T_{ref} \sim 288$ K, the term $k_r \ (1 - f_e)$ varies by $\sim 4.87$ Wm$^{-2}$ K$^{-1}$ to $\sim 0.54$ Wm$^{-2}$ K$^{-1}$ from dry ( $f_e$ =0) to wet ( $f_e$ =1) conditions.*"

Also, the following sentence is now added in the result section explaining that $\frac{dT'}{dR_s}$ is obtained by slope of the regression between warming rate and evaporative fraction, see L291-295

"*Mathematically, $\frac{dT'}{dR_s}$ is the change in warming rate from dry ( $f_e$ =0) to wet ( $f_e$ =1) condition which is obtained from the slope of the linear regression of daily warming rates and daily evaporative fraction.*"

11. **Figure 4: The label of the color bar is not correct.**

    Thank you. Corrected, see Figure R4 in the supplement of this response letter:

12. **L369: Figure 6a?**

    Corrected

13. **L373: same issue as above.**

    Corrected

**Supplement:**

Figure R1: The density distribution of the evaporative fraction for 51 FLUXNET sites for short vegetation, savanna and forests.

[Figure]

Figure R2: Warming rate response to evaporative fraction (a) and latent heat flux (b) for all the sites in short vegetation.

[Figure]

Figure R3: Demonstrating filter 1 that is used to select summer days for the analysis for US-ARM site.

The days with daily mean solar radiation greater than the median of the year are considered as summer days.

[Figure]

Figure R4: The legend of Figure 4 in the manuscript is corrected

[Figure]

---

## Author Comment (AC2) · 9 Jun 2020

**Response to the second reviewer**

Dear Reviewer,

Thank you for the thoughtful and thorough review of our manuscript. Your comments are helpful and we hope you will find our suggested revisions of this manuscript satisfactory.

Your comments are in blue bold color, which we answer in the black color. Some of

comprehensive responses are split, followed by your specific comments in blue. Text changes are highlighted in "*italic*" with the line numbers, and in red color within the manuscript. The line numbers might change in the final version of the manuscript. Figures related to this response letter are in the supplement and denoted by Figure Rn; here n is the figure number. Similarly changes in original figures are also present in the supplement and denoted as new Figure n.

**Interactive comments**

**This paper investigates the response of the diurnal warming rate of the surface and air temperature to evaporative conditions and vegetation cover type, which could be useful, as the authors point out, when estimating air temperatures from remote sensing of surface temperatures. They develop a simple model for the warming rate based on the surface energy balance which captures its observed response to ga and fe reasonably well. Overall, the idea is good and the study is thorough, so I recommend publication after revision of some issues. Some of these issues have already been addressed by my fellow referee.**

Thank you for these encouraging words. In order to address your concern we have gone through each of your suggestions.

1. **My main difficulty is to see why, when deriving equation (4), both the evaporative fraction and the aerodynamic conductance can be considered constant with Rs. This needs some justification. Later in the paper it is men-**

**tioned (L241-242) that evaporative fraction is stable during daylight hours, which should probably mentioned before presenting Eq. (4). Why can the diurnal variation of ga be ignored? The paper would in general benefit from a language revision.**

Thank you for these interesting observations.

The evaporative fraction is calculated from the slope of the linear regression of half hourly value of LE and LE+H during morning to noontime. We find that this ratio remains relatively constant, see Figure R1 in the supplement where we demonstrate examples of calculating evaporative fraction for a dry and a wet day. The assumption that the evaporative fraction remains constant during daytime is also supported by other literatures, which are already mentioned in the manuscript. To make it clear this is now mentioned before presenting Eq. (4) in the model section, see the updated text, L250-251.

"*Here we consider a daily constant morning to noon time evaporative fraction*"

Why can the diurnal variation of ga be ignored?

The authors would like to thank the reviewer for his thoughtful comments on role of diurnal aerodynamic conductance. We realize that this term has a significant impact in the model, which we have now considered and discussed in the manuscript. After this improvement most of the results remain similar, although slight improvement is now achieved in the model performance. This is reflected in new Figure 6 and new Figure 8, see the supplement.

This has lead to the following changes in the manuscript:

1. In the model section, considering diurnal $dg_a/dR_s$ , a new term ( $\frac{\overline{T_s-T_a}}{\overline{g_a}}$ . $\frac{dg_a}{dR_s}$ now adds to the Eq 5 such that the warming rate of surface temperature is given by

$$\frac{dT_s}{dR_s} \approx \frac{(1-f_e)}{c_p \ . \ \rho.\overline{g_a}} + \frac{dT_a}{dR_s} \ - \ \frac{\overline{T_s - T_a}}{\overline{g_a}} . \ \frac{dg_a}{dR_s}$$

   Here, $\overline{g_a}$ and $\overline{T_s - T_a}$ are the daily (morning time) mean aerodynamic conductance and surface and air temperature gradient, respectively.

2. Wherever the term daily mean aerodynamic conductance (before $g_a$ ) was used, is now replaced with $\overline{g_a}$ to avoid any confusion with its diurnal variation, which is now captured in term $dg_a/dR_s$ .

3. Eq 6 has been now removed from the model section and placed in the result section because the solution of Eq 6 requires the observed relationship of $\overline{T_s - T_a}$ to evaporative fraction, which is later shown in the result section in new Figure 5, see the supplement.

4. We realize that the model is more consistent when the aerodynamic conductance is calculated from the sensible heat flux than from frictional velocity and wind speed. This leads to some changes in Figure 5, which now demonstrates the density plot of $\overline{g_a}$ , $dg_a/dR_s$ and relationship of $\overline{T_s - T_a}$ to evaporative fraction.

5. Since we did not find a strong relationship between $\overline{g_a}$ (calculated from sensible heat flux) and $dg_a/dR_s$ to evaporative fraction (see Figure R2, Figure R3 in the supplement), we have now dropped out the related discussion from the text and from Figure 5 and previous Figure 7b.

6. Figure 6b and 7a are now merged to demonstrate the performance of the model in estimating warming rate and its response to evaporative fraction. The overall performance has slightly improved with the new version of model, see new Figure 6 in the supplement.

7. In addition to the effect of solar radiation, evaporative fraction and the mean aerodynamic conductance, Figure 7 (previously Figure 8) also discusses now the effect of $dg_a/dR_s$ on the diurnal surface temperature variation. This effect is mainly important for short vegetation and savanna. See new Figure 7 in the supplement.

The paper would in general benefit from a language revision.

For language revision the manuscript will go through the language copy editing at Copernicus.

**Some minor comments:**

1. **Plots using green and red are going to be difficult to read for colorblind people.**

   We agree with you. We have now changed green with blue and red with dark red. This change also leads to changes in the background colors of Figure 2.

2. **The dashed lines in Fig. 1 are not very easy to see**

   We have increased the width of the line to make it easily visible.

3. **In Fig. 5, may it be more useful to express the ga??? in the inset plot in relative terms (e.g. as a percentage of the mean aerodynamic conductance for each vegetation type)?**

   Thank you for this suggestion. However, with our new methodology of calculating $g_a$ from sensible heat flux and in accounting for the change with solar radiation, we found no strong relationship of aerodynamic conductance ($g_a$) to evaporative fraction; see Figure R2 in the supplement. In Figure 5, instead on $g_a'$ we have now added the density distribution of observed $dg_a/dR_s$ and $\overline{T_s - T_a}$ response to evaporative fraction which provide insight to the parameters needed to solve the model. See new Figure 5 in the supplement.

[Figure]

4. **L369 - Figure 6a**

   It has been now replaced with Figure 6b

5. **L406 - Figure 6 (a or b?)**

   It was Figure 6a but now it is removed from the manuscript.

6. **L445 - Where does this 74% come from?**

   This was the contribution of aerodynamic conductance in controlling the warming rate of surface temperature. However, in the new version of Figure 6 we do not need this.

**Some language and typos:**

1. **L73 "... the warming rate, (comma) that eliminates..."**

   Corrected

2. **L225 "Vegetation are classified into three types that is based on...". Rather "Vegetation IS classified into three types, based on..."?**

   Modified, see L237-238

"*Vegetation is classified into three types based on their typical height and coverage*"

3. **L251 - The year in bracketed citations shouldn???t be between brackets itself (see, e.g., Verma, 1989)**

Thank you for pointing it out.

4. **L427 - ".. contribution of the contribution..."**

This line is now removed after model modification.

5. **L487 - depend/depends**

Corrected.

6. **L533 - ambiguity? (uncertainty?)**

Corrected.

**Supplement:**

**Supplement for the response to #2 reviewer**

- **Modified figures in the manuscript (new)**

[Figure]

**New Figure 5** a) Observed density distribution of daily mean aerodynamic conductance $(\overline{g_a})$ during the time before noon. b) Observed density distribution of the sensitivity of aerodynamic conductance to solar radiation $dg_a/dR_s$ during the time before noon. c) Box plot of the mean surface and air temperature difference $(\overline{T_s - T_a})$ during the time before noon to evaporative fraction. The boxes indicate the 75th and 25th percentiles of the observations, respectively. The lines show the linear best fit for $\overline{T_s - T_a}$ to evaporative fraction for each vegetation type with the equations in the plot.

[Figure]

**New Figure 6** a) Modeled versus observed daily warming rates, $dT_s/dR_s$, for each site for the three vegetation types. The density distributions show the spread. The coefficient of determination ($r^2$) is depicted for the linear fit (dashed lines). b) Model evaluation of the response of surface temperature warming rates to evaporative conditions ($dT_s'/dR_s$) with those derived from observations for each site.

[Figure]

**New Figure 7 (old Figure 8)** Comparison of model estimates of the diurnal surface temperature range (DT$_s$R) for short vegetation, savanna and forests with observations

for four scenarios: a) DT$_s$R is only a function of solar radiation (R$_s$), b) DT$_s$R is a function of solar radiation (R$_s$) and evaporative fraction ($f_e$), c) DT$_s$R is a function of solar radiation and evaporative fraction, and mean aerodynamic conductance ($g_a$), and d) DT$_s$R is a function of solar radiation, evaporative fraction, aerodynamic conductance and diurnal variation of aerodynamic conductance ($dg_a/dR_s$). Dashed lines show the linear regression between model and observation.

- **Extra figures related to the response letter**

[Figure]

**Figure R1** Demonstrating the calculation of evaporative fraction ($f_e$) for a dry (a), and a wet (b) day of a cropland site in Southern Great Plains (US-ARM, FLUXNET). We use the slope of the linear regression of half hourly morning time observations of latent heat flux (LE) and total turbulent heat flux (LE +H) to obtain daily evaporative fraction.

[Figure]

**Figure R2** Boxplot of the variation of mean aerodynamic conductance with evaporative fraction for the three vegetation types.

[Figure]

**Figure R3** Boxplot showing the relationship between $dg_a/dR_s$ and evaporative fraction for the three vegetation types.

---

## Author Response (AR1)

**Letter to the Editor – Submission of revised manuscript**

Dear Dominic Mazvimavi,

Thank you for providing the opportunity to revise our manuscript. We are grateful to the reviewers for their constructive comments that have improved our manuscript, although the main findings remain same. We wish to submit the revised version of the manuscript for further consideration in the journal.

We would like to clarify the summary of the major changes adopted in the revised version, which are:

- There is slight change in the title of the manuscript. The new title is

   "Imprints of **evaporative conditions** and vegetation type in diurnal temperature variations"

   Following the suggestion of the first reviewer, we use now **evaporative conditions**, which is a more general term than **evaporation,** which could have been confused by the readers as the amount of water evaporated.

- An additional (new) affiliation of one of our coauthor (Maik Renner) is included.

- Following the concern of second reviewer, in Section 2 (2. Modeling temperature-warming rate), we have now considered the diurnal variation of aerodynamic conductance in our model; see Eq. (4) to Eq. (9) in the manuscript. The new variables are described within the text and also in Table A1 in the Appendix.

- In the new version, the aerodynamic conductance is calculated from sensible heat flux, a method that is consistent with our surface energy model. In the previous version, we used the frictional velocity for calculating the aerodynamic conductance in a way that is only suitable for neutral conditions and therefore not appropriate for the unstable conditions during daytime. This change is reflected in Figure 5, Table 2 and Figure A3 in the Appendix.

- With the new method to estimate aerodynamic conductance from the sensible heat flux, we did not find a relationship of aerodynamic conductance to evaporative fraction (see Figure RR1, in this response letter). Therefore, we have now removed the associated figures and text from the manuscript.

- To highlight our model performance, we have merged figure 6a and 7b from the old version of the manuscript into Figure 6 in the revised version.

- Figure 7 (previously Figure 8) is now reorganized in accordance to our updated model. Our conclusions remain nevertheless mostly the same. For a better interpretation of our findings we have calculated and discussed the contribution of each term contributing to the diurnal temperature range in percentages.

- The discussion on the implication of our findings and model is now extended. We have discussed how our study is valuable in understanding the diurnal temperature range; effects of land cover change, and the role of forests in cooling temperatures.

- The conclusion of our study remains similar to our previous version, now containing added and updated information following the suggestions of the reviewers.

- We improved the language, as recommended by Reviewer #2, and streamlined the text to reduce some redundancies.

To address all reviewers' comments, detailed responses are previously delivered. For completeness, these are added below, with the comments of reviewers in blue followed by our response in black. The manuscript text referred in the responses is shown in italics. In addition to these responses, we highlight all changes made in the manuscript in the text below, considering the major concerns and comments of the reviewers. The changes in the manuscript are highlighted in blue.

These additional documents cover the following page ranges:
- Page 3 to 4: Response to the First Reviewer
- Page 5 to 6: Response to the Second Reviewer
- Page 7 to : Marked up version of the revised manuscript.

We hope that we have addressed all of the comments from the reviewers in a satisfactory way.

We look forward to the outcome of your assessment.

Yours sincerely,
Annu Panwar
On behalf of the co-authors

**Response to the First Reviewer**

Dear Reviewer,

Thank you for finding our study interesting and relevant to the readers of the journal. We believe that we answered to your every comment in our previous response letter. Here we highlight all the changes stimulated by your review. Some of these changes are also discussed in response to the second reviewer.

- "Throughout the manuscript, "evaporation (or ET)" and "evaporative fraction" are used as interchangeable terms."

  For avoiding such confusion we use term "evaporative conditions" and "evaporative fraction" throughout the manuscript.

  This is also reflected in the title, which was,

  *"Imprints of evaporation and vegetation type in diurnal temperature variations"*

  And now is modified to,

  *"Imprints of evaporative condition and vegetation type in diurnal temperature variations"*

- The authors should note that aerodynamic conductance and evaporative fraction are not independent (Rigden and Li 2017). However, in their analysis, aerodynamic conductance and evaporative fraction were treated as two independent factors that govern the diurnal variations of surface temperature.

  We would like to notify that in the revised version of the manuscript, our calculation of aerodynamic conductance is based on sensible heat flux and not on frictional velocity. Calculating aerodynamic conductance from sensible heat flux makes our surface energy balance model consistent. This change has slightly improved our model performance and its interpretation.

  Based on our updated analysis of aerodynamic conductance we find that the aerodynamic conductance and evaporative fraction has no strong relationship in the observations. See below, Figure RR1.

[Figure]

**Figure RR1** Box plot of observed relationship between morning mean aerodynamic conductance and morning mean evaporative fraction in short vegetation, savanna and forest.

These results are in contradiction to Rigden and Li 2017, because similar to our previous method they calculate aerodynamic conductance from frictional velocity and not from sensible heat flux. This is a very important distinction that we also highlight in the manuscript. See lines 379-382,

*"This finding is different to the study by Rigden and Li (2017) who showed that the aerodynamic resistance depends on the Bowen ratio. This difference can be attributed to their way to estimate aerodynamic resistance from the frictional velocity and wind speeds, which assumes neutral conditions, whereas we obtain aerodynamic conductance from sensible heat flux."*

- One of the main conclusions of this study is that the warming rates do not carry imprints of evaporation in the forest. What is the physical mechanism of this absent imprints? Why does the high aerodynamic conductance of the forest result in no imprint of evaporation in diurnal temperature variations? Does this necessarily mean ET of the forest has little impact on temperature? The authors need to provide more explanations of the physical processes and more discussions on the implications of the warming rate sensitivity.

This is very intriguing observation.

For clear interpretation of our results we have now added the contribution (in %) of evaporative conditions and aerodynamic conductance in cooling temperatures, see lines 490 to 497,

*"Evaporation reduces $DT_sR$ by ~18 % (short vegetation: (1.26-1.51)/1.51 = -16.55 %, savanna: (1.37 – 1.64)/1.64 = -16.46 %, and forest: (2.22 – 2.84)/2.84 = -21.83 %). On comparing Figure 7 b and 7c, we found that the high aerodynamic conductance of forests reduces $DT_sR$ by 56 % ((0.98 – 2.22)/2.22 = -56%). In other words, the higher aerodynamic conductance of forests causes a substantially larger cooling than evaporation. The diurnal variation of the aerodynamic conductance then reduces the $DT_sR$ further, being stronger in short vegetation ((0.86 – 1.32)/1.32 = -35%) and savanna ((0.90 – 1.40)/1.75 = -35.25 %) than in forests ((0.77 – 0.98)/0.98= -21%)."*

Our model and observational findings shows that it is the high aerodynamic conductance of forest that cools the diurnal temperature; evaporative cooling is a secondary factor. This is an important finding, and is now discussed in detail in the discussion section. See lines 561 to 567,

*"… the low $DT_sR$ of forests mostly to their high aerodynamic conductance (~56 %), with evaporation playing only a secondary role (~22 %). This finding is consistent with studies that showed that the warming induced by deforestation to be mainly the consequence of changes in aerodynamic conductance rather than changes in evaporative conditions (Bright et al., 2017; Chen and Dirmeyer, 2016; Lee et al., 2011b; Zhao et al., 2014b). This aerodynamic effect is thus important for the cooling effect of forests (Ellison et al., 2017; Li et al., 2015; Tang et al., 2018), which our analysis and analytical model supports."*

Does this necessarily mean ET of the forest has little impact on temperature? What is the physical mechanism of this absent imprints?

Yes, our study shows that the aerodynamic conductance of the forest is the dominant cooling agent whereas evaporation is secondary. The mechanism is discussed in discussion section in lines 550 to 558,

*"We can only speculate about the physical mechanism behind this finding. While it is well established that the greater aerodynamic roughness of the forest leads to a greater aerodynamic conductance for neutral conditions (T.R. Oke, 2002) we also find that the diurnal variation is much larger than the mean (the term $dg_a/dR_s$). This enhancement is most likely related to buoyancy, which is produced when the surface is heated by the absorption of solar radiation during the day. The finding that the relative enhancement of aerodynamic conductance between forests and non-forests is the same, and that this enhancement is insensitive to evaporative fraction, seems to be surprising and would need further investigations about their physical explanations."*

Thank you

**Response to the Second Reviewer**

Dear Reviewer,

Thank you for your constructive comments and encouraging words. Your suggestion to consider diurnal cycle of aerodynamic conductance has improved our model and its interpretation. The main results remain same but with your suggestion a broader prospective on the importance of aerodynamic conductance is now recognized in the revised manuscript. Here we revisit your comments that are implemented and discussed in the manuscript.

- My main difficulty is to see why, when deriving equation (4), both the evaporative fraction and the aerodynamic conductance can be considered constant with Rs. This needs some justification. Later in the paper it is mentioned (L241-242) that evaporative fraction is stable during daylight hours, which should probably mentioned before presenting Eq. (4). Why can the diurnal variation of ga be ignored?

   Thank you for pointing out this. In the revised version, diurnal aerodynamic conductance, represented as $dg_a/dR_s$, is now considered in the model, and in our observational analysis. Following are the changes resulted on using $dg_a/dR_s$,

1. In the model section the warming rate is now given by (Eq. 7), Line 198

$$\frac{dT_s}{dR_s} \approx \frac{(1-f_e)}{c_p \cdot \rho \cdot \overline{g_a}} + \frac{dT_a}{dR_s} - \frac{\overline{T_s - T_a}}{\overline{g_a}} \cdot \frac{dg_a}{dR_s}$$

, And its response to evaporative fraction by (Eq. 9), Line 213

$$\left(\frac{dT_s}{dR_s}\right)' = -\frac{1}{c_p \cdot \rho \cdot \overline{g_a}} + \frac{dT_a}{dR_s}' - \left[\frac{\overline{(T_s - T_a)}}{\overline{g_a}} \cdot \frac{dg_a}{dR_s}\right]'$$

   The last term of Eq.9 is solved using observations, shown in Figure 5 and discussed in lines 373 to 388.

   We have now emphasis more on demonstrating the implication of our study in estimating and analyzing factors shaping the diurnal range of surface temperature ($DT_sR$), which is a known index for quantifying the diurnal variation of temperatures.

2. In section 3 (3. Data and Methods), we mention the updated method of calculating aerodynamic conductance and its diurnal variation. See lines 278 to 282

   *"The aerodynamic conductance ( $g_a$) is obtained from the observed sensible heat flux from $g_a=H/(c_p \cdot \rho \cdot (T_s - T_a))$. Since aerodynamic conductance is not constant over the day, its diurnal variation is described by $dg_a/dR_s$, which is estimated from a linear regression of morning to noon half hourly values of $g_a$ and $R_s$."*

3. In section 4 (4.1 observational analysis), we have added the information on aerodynamic conductance and its diurnal variation in Table 2; which is also demonstrated in Figure 5.

4. We have also updated the representation of model performance in Figure 6, where we show how well the model reproduces the warming rate and its response to evaporative fraction. The overall performance of model has slightly improved after considering the contribution of diurnal aerodynamic conductance.

5. Figure 7 demonstrates the contribution of each term in cooling diurnal range of surface temperature. In the revised version we also distinguish the contribution of mean aerodynamic conductance and its diurnal variation. For better interpretation, we have summarized the contributions in percentage, See Lines 488 to 497.

6. We realize that our findings and their implications require more discussion, that is why we have added two new paragraphs in the discussion section, see Lines 545-558 and 569 to 577.

7. Conclusions of our studies remain almost the same. However, we have added the following lines to demonstrate the importance of diurnal variation of aerodynamic conductance and its role in cooling diurnal temperatures.

Line 609 to 613

*".. in forests, surface temperature is strongly aerodynamically coupled to air temperatures by their high aerodynamic conductance, so that these lack sensitivity to evaporative fraction. Hence, diurnal temperature variations of forested sites do not seem to carry a notable effect of evaporation. What this shows is that the effect of evaporative conditions on diurnal temperature variations delicately depends on the presence or absence of forests.*
*And Line 658 to 661"*

8. We have updated figure A3 that shows the mean morning time aerodynamic conductance of each site.

• Plots using green and red are going to be difficult to read for colorblind people.

In this version of the manuscript we use dark red, grey and blue color for short vegetation, savanna and forests, respectively. In addition to distinguishing colors, we also use different symbols in scatter plots.

[revised manuscript text omitted]